# CAN TRANSFORMERS IN-CONTEXT LEARN BEHAVIOR OF A LINEAR DYNAMICAL SYSTEM?

## ABSTRACT

We investigate whether transformers can learn to track a random process when given observations of a related process and parameters of the dynamical system that relates them as context. More specifically, we consider a finite-dimensional state-space model described by the state transition matrix $F$, measurement matrices $h_1, \ldots, h_N$, and the process and measurement noise covariance matrices $Q$ and $R$, respectively; these parameters, randomly sampled, are provided to the transformer along with the observations $y_1, \ldots, y_N$ generated by the corresponding linear dynamical system. We argue that in such settings transformers learn to approximate the celebrated Kalman filter, and empirically verify this both for the task of estimating hidden states $\hat{x}_{N|1,2,3,\ldots,N}$ as well as for one-step prediction of the $(N+1)^{st}$ observation, $\hat{y}_{N+1|1,2,3,\ldots,N}$. A further study of the transformer's robustness reveals that its performance is retained even if the model's parameters are partially withheld. In particular, we demonstrate that the transformer remains accurate at the considered task even in the absence of state transition and noise covariance matrices, effectively emulating operations of the Dual-Kalman filter.

## 1 INTRODUCTION

In-context learning, in particular few-shot prompting (Yogatama et al., 2019), is a growing area of research in natural language processing (NLP). In this framework, a large language model (LLM) learns tasks from a relatively few examples, i.e., few demonstrations of input-output pairs. One of the earliest works to show that LLMs are capable of being fine-tuned when provided with prompts was by Brown et al. (2020); there, the authors evaluated the GPT-3 model over a plethora of NLP datasets and various "zero-shot", "one-shot", and "few-shot" learning tasks. In (Zhao et al., 2021), the authors implicate majority label bias, recency bias, and common token bias as the reasons for instability in GPT-3's accuracy following few-shot prompting and propose a contextual calibration procedure as a remedy. A theoretical analysis proving that language models perform implicit Bayesian inference is presented in (Xie et al., 2021). Min et al. (2022) explore the reasons why in-context learning works, show that in-context learning is not affected by the lack of ground-truth labels, and posit that the label space and the distribution of the input text along with the format of the prompts play a crucial role. Moreover, Schlag et al. (2021) theoretically show that transformers are fast weight programmers.

Early works that explore using standard transformer decoders to in-context learn auto-regressive models include Garg et al. (2022); there, the authors empirically investigate the ability of transformers to learn classes of linear functions. They say that a model learns a function class $\mathcal{F}$ with domain $\mathcal{X}$ if for any $f \in \mathcal{F}$ and for any $x_1, x_2, \ldots, x_N, x_{query}$ sampled from $\mathcal{X}$ in an IID fashion, the model is able to predict the output $f(x_{query})$ given the sequence $x_1, f(x_1), x_2, f(x_2), \ldots, x_N, f(x_N), x_{query}$. The classes explored in Garg et al. (2022) range from simple linear functions to sparse linear functions, two-layered neural networks, and decision trees. Two parallel works, Von Oswald et al. (2023) and Akyürek et al. (2023), explored which algorithms does the transformer resemble the most as it learns the functional classes in-context. In Von Oswald et al. (2023), the authors build on the work of Schlag et al. (2021) to elegantly show that the transformations induced by linear self-attention can be perceived as equivalent to a gradient descent step. In other words, for a single 1-head linear self attention layer there exist key, query, and value matrices such that a forward pass on the transformer resembles the execution of one step of gradient descent with L2 loss on every token. Akyürek et al. (2023) take a fundamentally different approach, defining a raw operator that can be used to perform

various operations on the input tokens including matrix multiplication, scalar division, and read-write; they then show that a single transformer head with appropriate key, query, and value matrices can approximate the raw operator. This implies that by using the operations readily implemented by the raw operator, transformers are in principle capable of implementing linear regression via stochastic gradient descent or closed-form regression.

In this work, we investigate whether in-context learning may enable a transformer to predict states and/or outputs of a linear dynamical system described by a state-space model with non-scalar state transition matrix, non-zero process noise, and white measurement noise. For such systems, Kalman filter (Kalman, 1960) is the optimal (in the mean-square error sense) linear state estimator. We investigate what algorithm the transformer most closely resembles as it learns to perform one-step prediction when provided context in the form of observations generated by a system with arbitrarily sampled state transition matrix, time-varying measurement matrices, and the process and observation noise covariance matrices. We show that Kalman filtering can be expressed in terms of operations readily approximated by a transformer; this implies that when given the observations and system parameters as context, the transformers can in principle emulate the Kalman filtering algorithm. This is corroborated by extensive experimental results which demonstrate that such in-context learning leads to behavior closely mimicking the Kalman filter when the context lengths are sufficiently large. Interestingly, the transformer appears capable of emulating the Kalman filter even if some of the parameters are withheld from the provided context, suggesting robustness and potential ability to implicitly learn those parameters from the remaining context.

Prior works that investigate interplay between deep learning and Kalman filtering notably include Deep Kalman Filters Krishnan et al. (2015) and Kalman Nets (Revach et al., 2021); the latter is a framework that circumvents the need for accurate estimates of system parameters by learning the Kalman gain in a data-driven fashion using a recurrent neural network. The follow-up work (Revach et al., 2022) employs gated recurrent units to estimate the Kalman gain and noise statistics while training and evaluating, as in (Revach et al., 2021), the proposed model on data generated by a system with fixed parameters. In contrast, in our work the model parameters are randomly sampled to generate each training example, leading transformer to learn how to perform filtering rather than memorize input-output relationship of a specific system. Dao & Gu (2024) study the theoretical connections between structured state space models (SSMs) and variants of attentions. The central message of their work is that the computations of various SSMs can be re-expressed as matrix multiplication algorithms on structured matrices, an insight that can be utilized to show the relationship between selective SSMs and attention to make the latter efficient. Sieber et al. (2024) introduce a dynamical systems framework (DSF) to find a common representation unifying attention, SSMs, Recurrent Neural Networks (RNNs) and LSTMs. However, these works bear no relevance to the problem of state estimation, filtering, or in-context learning in general.Goel & Bartlett (2024) show that softmax self attention can represent Nadarya-Watson smoothing estimator, and proceed to argue that this estimator approximates Kalman filter. In contrast, we explicitly focus on the problem of in-context learning and build on the concepts proposed by Akyürek et al. (2023) to show that transformers implement exact operations needed to perform Kalman filtering, supporting these arguments with extensive empirical results. To the best of our knowledge, the current paper reports the first study of the ability of transformers to in-context learn to emulate Kalman filter using examples generated by randomly sampling parameters of an underlying dynamical system.

The remainder of this paper is organized as follows. Section 2 provides an overview of relevant background. Section 3 lays out the system model and presents theoretical arguments that transformers can in-context learn to implement Kalman filtering for white observation noise. Section 4 reports the simulation results, including empirical studies of the robustness to missing model parameters, while Section 5 concludes the paper.

## 2 BACKGROUND

### 2.1 TRANSFORMERS

Transformers, introduced byVaswani et al. (2017), are neural networks architectures that utilize the so-called attention mechanism to map an input sequence to an output sequence. Attention mechanism facilitates learning the relationship between tokens representing the input sequence, and is a

key to the success of transformers in sequence-to-sequence modeling tasks. The experiments in this paper utilize the GPT2-based (decoder-only) architecture Radford et al. (2019).

A brief overview of the attention mechanism will help set the stage for the upcoming discussion. Let $G^{(l-1)}$ denote the input of the $l^{th}$ layer. A single transformer head, denoted by $\gamma$, consists of key, query, and value matrices denoted by $W_\gamma^K$, $W_\gamma^Q$, and $W_\gamma^V$, respectively. The output of the head $\gamma$ is computed as

$$b_\gamma^l = \text{Softmax}\left((W_\gamma^Q G^{(l-1)})^T (W_\gamma^K G^{(l-1)})\right)\left(W_\gamma^V G^{(l-1)}\right). \tag{1}$$

The softmax term in equation 1, informally stated, assigns weights to how tokens at two positions are related to each other. The output of all the $B$ heads are concatenated and combined using $W^F$ to form

$$A^l = W^F[b_1^l, b_2^l, ..., b_B^l]. \tag{2}$$

The resulting output is then passed to the feedforward part of the transformer block to obtain

$$G^{(l)} = W_1\sigma\left(W_2\lambda\left(A^l + G^{(l-1)}\right)\right) + A^l + G^{(l-1)}, \tag{3}$$

where $\sigma$ denotes the non-linear activation function and $\lambda$ denotes layer normalization. For our experiments, we use Gaussian error linear unit (GeLU Hendrycks & Gimpel (2016)) as the activation function.

## 2.2 IN-CONTEXT LEARNING FOR LINEAR REGRESSION

Let us consider linear dynamical systems described by a finite-dimensional state-space model involving hidden states $x_t \in R^n$ and observations (i.e., measurements) $y_t \in R^m$ related through the system of equations

$$x_{t+1} = F_t x_t + q_t \tag{4}$$
$$y_t = H_t x_t + r_t. \tag{5}$$

The state equation (4), parameterized by the state transition matrix $F_t \in \mathbb{R}^{n \times n}$ and the covariance $Q$ of the stationary zero-mean white process noise $q_t \in \mathbb{R}^n$, captures the temporal evolution of the state vector. The measurement equation (5), parameterized by the measurement matrix $H_t \in \mathbb{R}^{m \times n}$ and the covariance $R$ of the stationary zero-mean white measurement noise $r_t \in R^m$, specifies the acquisition of observations $y_t$ via linear transformation of states $x_t$. Such state-space models have proved invaluable in machine learning (Gu et al., 2021), computational neuroscience (Barbieri et al., 2004), control theory (Kailath, 1980), signal processing (Kailath et al., 2000), economics (Zeng & Wu, 2013), and other fields. Many applications across these fields are concerned with learning the hidden states $x_t$ given the noisy observations $y_t$ and the parameters of the state space model.

Assume a simple setting where $F = I_{n \times n}$, $Q = 0$, $H = h_t \in \mathbb{R}^{1 \times n}$ (i.e., scalar measurements), and $x_0 = x$. Here, the state space model simplifies to

$$x_t = x \tag{6}$$
$$y_t = h_t x_t + r_t, \tag{7}$$

i.e., the state equation becomes trivial and the system boils down to a linear measurement model in equation (7). In this setting, inference of the unknown random vector $x$ given the observations $y_1, y_2, ..., y_N$ and the measurements vectors $h_1, h_2, h_3, ..., h_N$ is an estimation problem that can readily be solved using any of several well-known techniques including:

- **Stochastic Gradient Descent.** After initializing it as $\hat{x}_0 = \mathbf{0}_{\mathbf{n \times 1}}$, the state estimate is iteratively updated by going through the measurements and recursively computing

$$\hat{x}_t = \hat{x}_{t-1} - 2\alpha(h_{t-1}\hat{x}_{t-1}^T h_{t-1} - h_{t-1}y_{t-1}), \tag{8}$$

    where $\alpha$ denotes the learning rate. Once a pre-specified convergence criterion is met, the final estimate is set to $\hat{x}_{SGD} = \hat{x}_N$.

- **Ordinary Least Squares (OLS).** Let the matrix $\bar{\mathbf{H}} \in \mathbb{R}^{N \times n}$ be such that its rows are measurement vectors, i.e., the $i-th$ row of $\bar{\mathbf{H}}$ is $h_i$; furthermore, let $\bar{\mathbf{Y}} = [y_1, y_2, ..., y_N]^T$. Then the OLS estimator is found as

$$\hat{x}_{OLS} = (\bar{\mathbf{H}}^T \bar{\mathbf{H}})^{-1} \bar{\mathbf{H}}^T \bar{\mathbf{Y}}. \tag{9}$$

- **Ridge Regression.** To combat overfitting and promote generalization, the ridge regression estimator regularizes the OLS solution as

$$\hat{x}_{Ridge} = (\bar{\mathbf{H}}^T \bar{\mathbf{H}} + \lambda I_{n \times n})^{-1} \bar{\mathbf{H}}^T \bar{\mathbf{Y}}, \tag{10}$$

where $\lambda$ denotes the regularization coefficient.

It is worth pointing out that if $\lambda = \frac{\sigma^2}{\tau^2}$, where $\sigma^2$ is the variance of $r_t$ and $\tau^2$ is the variance of $x_0 = x$, ridge regression yields the lowest mean square error among all linear estimators of $x$, i.e., the estimators that linearly combine measurements $y_1, ..., y_N$ to form $\hat{x}$. Furthermore, if $x_0 \sim \mathcal{N}(0, \tau^2 I)$ and $r_t \sim \mathcal{N}(0, \sigma^2 I)$, the ridge regressor yields the minimum mean square error estimate that coincides with $E[X|y_1, ..., y_N]$.

A pioneering work that explored the capability of language models to learn linear functions and implement simple algorithms was reported by Garg et al. (2022). The ability of a transformer to learn $x_t = x$ in (7) was studied by Akyürek et al. (2023) which, building upon (Garg et al., 2022), explored what algorithms do GPT-2 based transformers learn to implement when trained in-context to predict $y_N$ given the input organized into matrix

$$\begin{bmatrix} 0 & y_1 & 0 & y_2 & ... & 0 & y_{N-1} & 0 \\ h_1^T & 0 & h_2^T & 0 & ... & h_{N-1}^T & 0 & h_N^T \end{bmatrix}.$$

Training the transformer in (Akyürek et al., 2023) was performed by utilizing data batches comprising examples that consist of randomly sampled states and parameters. It was argued there that for limited architecture models trained on examples with small context lengths, the transformer approximates the behavior of the stochastic gradient descent algorithm. For moderate context lengths less than or equal to the state dimension and moderately sized model architectures, the transformer mimics the behavior of Ridge Regression; finally, for context lengths greater than the state dimensions and large transformer models, the transformer matches the performance of Ordinary Least Squares.

A major contribution of Akyürek et al. (2023) was to theoretically show that transformers can approximate the operations necessary to implement SGD or closed-form regression. This was accomplished by introducing and utilizing the *RAW* (Read–Arithmetic–Write) operator parameterized by $W_o, W_a, W$ and the element-wise operator $\circ \in [+, *]$ that maps the input to the layer $l$, $\mathbf{q}^l$, to the output $\mathbf{q}^{l+1}$ for the index sets $s, r, w$, time set map $K$, and positions $i = 1, \ldots, 2N$ according to

$$q_{i,w}^{l+1} = W_o \left( \left[ \frac{W_a}{|K(i)|} \sum_{k \in K(i)} q_k^l[r] \right] \circ W q_i^l[s] \right), \tag{11}$$

$$q_{i,j \notin w}^{l+1} = q_{i,j \notin w}^l. \tag{12}$$

A single transformer head can approximate this operator for any $W_o, W_a, W$, and $\circ$; moreover, there exist $W_o, W_a, W, \circ \in \{+, *\}$ that approximate operations necessary to implement SGD and closed-form regression including affine transformations, matrix multiplications, scalar division, dot products, and read-write operations.

## 3 IN-CONTEXT LEARNING FOR FILTERING AND PREDICTION OF A DYNAMICAL SYSTEM

Here we outline an in-context learning procedure for the generic state-space model given in (4)-(5), where we assume time-invariant state equation (i.e., $F_t = F \neq I$, $Q \neq 0$). For the simplicity of presentation, we at first consider scalar measurements. In such settings, the causal linear estimator of the state sequence $x_t$ that achieves the lowest mean-square error is given by the celebrated Kalman filter (Kalman (1960)). Specifically, one first sets the estimate and the corresponding error

covariance matrix of the initial state to $\hat{x}_0^+$ and $\hat{P}_0^+$, respectively. For our work, we let $\hat{x}_0^+ = 0$ and $\hat{P}_0^+ = I_{n \times n}$. Then the estimates and the corresponding error covariance matrices of the subsequent states are found recursively via the prediction and update equations of the Kalman filter as stated below.

**Prediction Step**:

$$\hat{x}_t^- = F\hat{x}_{t-1}^+ \tag{13}$$

$$\hat{P}_t^- = F\hat{P}_{t-1}^+ F^T + Q \tag{14}$$

**Update Step**:

$$K_t = \hat{P}_t^- H_t^T (H_t \hat{P}_t^- H_t^T + R)^{-1} \tag{15}$$

$$\hat{x}_t^+ = \hat{x}_t^- + K_t(y_t - H_t \hat{x}_t^-) \tag{16}$$

$$\hat{P}_t^+ = (I - K_t H_t)\hat{P}_t^- \tag{17}$$

For scalar measurements $H_t = h_t$ (a row vector) and $R = \sigma^2$ (a scalar), simplifying the computationally intensive matrix inversion in (15) into simple scalar division readily approximated by a transformer head. Then the update equations become

$$\hat{x}_t^+ = \hat{x}_t^- + \frac{\hat{P}_t^- h_t^T}{h_t \hat{P}_t^- h_t^T + \sigma^2}(y_t - h_t \hat{x}_t^-) \tag{18}$$

$$\hat{P}_t^+ = (I - \frac{\hat{P}_t^- h_t^T h_t}{h_t \hat{P}_t^- h_t^T + \sigma^2})\hat{P}_t^-, \tag{19}$$

involving operations that, as argued by Akyürek et al. (2023), are readily implemented by transformers. To investigate how closely can a transformer mimic the behavior of the Kalman filter when trained through in-context learning, we provide it with generic examples consisting of randomly generated $F, h_1, ..., h_N, \sigma^2$ and $Q$ structured as the $(n+1) \times (2n+2N+1)$ matrix

$$\begin{bmatrix} 0 & 0 & \sigma^2 & 0 & y_1 & 0 & y_2 & ... & y_{N-1} & 0 \\ F & Q & 0 & h_1^T & 0 & h_2^T & 0 & ... & 0 & h_N^T \end{bmatrix}. \tag{20}$$

The transformer, whose output is denoted by $T_\theta()$, can then be trained against the output at every second position starting from the position $2n+1$, with the loss function

$$\frac{1}{N}\sum_{t=1}^{N}(y_t - T_\theta(h_1, y_1, ..., h_{t-1}, y_{t-1}, h_t, F, Q, \sigma^2))^2. \tag{21}$$

Recall that, as shown in (Akyürek et al., 2023), there exist a parametrization of a transformer head that can approximate the operator in (11)-(12). Below we specify operations, readily implemented using the operator (11)-(12), which can be used to re-state the Kalman filtering prediction and update steps. These operations are defined on the subsets of indices of the input matrix. As an illustration of such a subset, let us consider matrix $F$; the set of indices specifying position of $F$ in expression (67) is given by $I_F^{input} = [(1,0),(1,1),(1,2),...,(1,n-1),...,(n,0),(n,1),...,(n,n-1)]$. Further details of such a construction are provided in the appendix. We define the operations needed to re-state the Kalman filtering steps as follows:

1. **Mul**$(I, J, K)$. The transformer multiplies the matrix formed by the entries corresponding to the indices in set $I$ with the matrix formed by the entries corresponding to the indices in set $J$, and writes the result on the indices specified by the set $K$.

2. **Div**$(I, j, K)$. The transformer divides the entries corresponding to the indices in set $I$ by the scalar at the coordinate $j$ and stores the result at the indices specified by the set $K$.

3. **Aff**$(I, J, K, W_1, W_2)$. This operation implements the following affine transformation: The transformer multiplies the matrix formed by the entries corresponding to the indices in set $I$ with $W_1$ and adds it to $W_2$ multiplied by the matrix formed by the entries corresponding to the indices in set $J$; finally, the result is written on the indices specified by the set $K$.

4. **Transpose**$(I, J)$. This operation finds the transpose of the matrix at $I$ and writes it to $J$.

It is straightforward to re-state the Kalman filtering recursions using the operations specified above. However, to do so, we first require some additional notation. We assume that a matrix consisting of zero and identity submatrices may be prepended to the input to the transformer. Let us denote the prepended matrix by $\mathcal{A}_{append}$, and let the resulting matrix be $\mathcal{A}_{cat} = [\mathcal{A}_{append}, \mathcal{A}_{input}]$. We denote by $I_{B1}$ the index set pointing to an $n \times n$ identity submatrix in $\mathcal{A}_{cat}$. Moreover, let $I_{B2}$ and $I_{B9}$ denote indices of two $n \times n$ submatrices of zeros in $\mathcal{A}_{cat}$; let $I_{B3}$ denote indices of a $1 \times n$ submatrix of zeros; let $I_{B4}$ and $I_{B8}$ denote indices of two $n \times 1$ submatrices of zeros; and let $I_{B5}$, $I_{B6}$, and $I_{B7}$ denote indices of (scalar) zeros in $\mathcal{A}_{cat}$. Finally, let the index sets of $F$, $Q$, and $\sigma^2$ in $\mathcal{A}_{cat}$ be denoted by $I_F$, $I_Q$, $I_\sigma$ respectively. With this notation in place, the Kalman filtering recursion can be formally restated as Algorithm 1. Note that all this additional notation introduced above concerns initializations and buffers to write the variables into. By concatenating them to the input matrix, we simply create convenient space to write the state, state covariance and other intermediate variables, ultimately arriving at Algorithm 1.

The presented framework is generalizable to non-scalar measurements with IID noise. To see this, suppose $y_t \in R^m$ and $r_t \sim \mathcal{N}(0, R)$, where $R$ is a diagonal $m \times m$ positive definitive matrix with diagonal entries $\sigma_1^2, \sigma_2^2, \ldots, \sigma_m^2$. Let $H_t$ denote the measurement matrix at time step $t$. Furthermore, let $y_t^j$ denote the $j^{th}$ component of $y_t$, and let $H_t^{(j)}$ denote the $j^{th}$ row of $H_t$. The Kalman filter recursions then become (Kailath et al., 2000)

$$\hat{x}_t^{(1)+} = \hat{x}_t^- + \frac{\hat{P}_t^- H_t^{(1)T}}{H_t^{(1)} \hat{P}_t^- H_t^{(1)T} + \sigma_1^2}(y_t^{(1)} - H_t^{(1)T} \hat{x}_t^-) \tag{22}$$

$$\hat{P}_t^{(1)+} = (I - \frac{\hat{P}_t^- H_t^{(1)T} H_t^{(1)}}{H_t^{(1)} \hat{P}_t^- H_t^{(1)T} + \sigma_1^2})\hat{P}_t^- \tag{23}$$

$$\hat{x}_t^{(j)+} = \hat{x}_t^{(j-1)+} + \frac{\hat{P}_t^{(j-1)+} H_t^{(j)T}}{H_t^{(j)} \hat{P}_t^{(j-1)+} H_t^{(j)T} + \sigma_j^2}(y_t^{(j)} - H_t^{(j)T} \hat{x}_t^{(j-1)+}) \quad j = 2, \ldots, m \tag{24}$$

$$\hat{P}_t^{(j)+} = (I - \frac{\hat{P}_t^{(j-1)+} H_t^{(j)T} H_t^{(j)}}{H_t^{(j)} \hat{P}_t^{(j-1)+} H_t^{(j)T} + \sigma_j^2})\hat{P}_t^{(j-1)+} \quad j = 2, \ldots, m \tag{25}$$

$$\hat{x}_t^+ = \hat{x}_t^{(m)+} \tag{26}$$

$$\hat{P}_t^+ = \hat{P}_t^{(m)+} \tag{27}$$

The in-context learning can be performed by providing to the transformer the input formatted as

$$\begin{bmatrix} 0 & 0 & \sigma_1^2 & 0 & y_1^{(1)} & \ldots & 0 & y_{N-1}^{(1)} & 0 \\ 0 & 0 & \sigma_2^2 & 0 & y_1^{(2)} & \ldots & 0 & y_{N-1}^{(2)} & 0 \\ . & . & . & . & . & . & . & . & . \\ . & . & . & . & . & . & . & . & . \\ 0 & 0 & \sigma_m^2 & 0 & y_1^{(m)} & \ldots & 0 & y_{N-1}^{(m)} & 0 \\ F & Q & 0 & H_1^{(1)T} & 0 & \ldots & H_{N-1}^{(1)T} & 0 & H_N^{(1)T} \\ . & . & . & . & . & . & . & . & . \\ 0 & 0 & 0 & H_1^{(m)T} & 0 & \ldots & H_{N-1}^{(m)T} & 0 & H_N^{(m)T} \end{bmatrix}, \tag{28}$$

allowing a straightforward extension of the algorithm to the non-scalar measurements case (omitted for the sake of brevity).

## 4 SIMULATION RESULTS

For transparency and reproducibility, we build upon the code and the model released by Garg et al. (2022). All the results are obtained on a 16-layered transformer model with 4 heads and hidden size 512. We implement curriculum learning initialized with context length $N = 10$, incremented by 2 every 2000 training steps until reaching the context length of 40. The dimension of the hidden state in all experiments was set to $n = 8$. Every training step is performed using Adam optimizer Kingma (2014) with a learning rate of 0.0001 on a batch of 64 examples, where for each example

**Algorithm 1** *Formulating the Kalman filter recursions using the elementary operations implementable by transformers.*

1: **Input**: $\mathcal{A}_{cat}, I_F, I_Q, I_\sigma, I_{B1}, I_{B2}, I_{B3}, I_{B4}, I_{B5}, I_{B6}, I_{B7}, I_{B8}, I_{B9}$
2: **Initialize** $I_{\hat{X}_{Curr}} \leftarrow (1:n, 2n)$
3: **for** $i = 1$ to $N$ **do**
4:    $I_{\hat{X}_{next}} \leftarrow (1:n, 2n + 2i)$
5:    $I_h \leftarrow (1:n, 2n + 2i - 1)$
6:    $I_y \leftarrow (0, 2n + 2i)$
7:    **Transpose**($I_F, I_{B2}$)
8:    **Mul**($I_F, I_{\hat{X}_{Curr}}, I_{\hat{X}_{next}}$)
9:    **Mul**($I_F, I_{B1}, I_{B1}$)
10:    **Mul**($I_{B1}, I_{B2}, I_{B1}$)
11:    **Aff**($I_{B1}, I_Q, I_{B1}, W_1 = I_{n \times n}, W_2 = I_{n \times n}$)
12:    **Transpose**($I_h, I_{B3}$)
13:    **Mul**($I_{B1}, I_h, I_{B4}$)
14:    **Mul**($I_{B3}, I_{B4}, I_{B5}$)
15:    **Aff**($I_{B5}, I_\sigma, I_{B6}, W_1 = 1, W_2 = 1$)
16:    **Div**($I_{B4}, I_{B6}, I_{B4}$)
17:    **Mul**($I_h, I_{\hat{X}_{next}}, I_{B7}$)
18:    **Aff**($I_y, I_{B7}, I_{B7}, W_1 = 1, W_2 = -1$)
19:    **Mul**($I_{B7}, I_{B4}, I_{B8}$)
20:    **Aff**($I_{\hat{X}_{next}}, I_{B8}, I_{\hat{X}_{next}}, W_1 = 1, W_2 = 1$)
21:    **Mul**($I_{B4}, I_{B3}, I_{B9}$)
22:    **Mul**($I_{B9}, I_{B1}, I_{B9}$)
23:    **Aff**($I_{B1}, I_{B9}, I_{B1}, W_1 = I_{n \times n}, W_2 = -I_{n \times n}$)
24:    $I_{\hat{X}_{Curr}} \leftarrow I_{\hat{X}_{next}}$
25: **end for**

$x_0, H_1, H_2, \ldots, H_N$ are sampled from isotropic Gaussian distributions. The process noise $q_t$ is sampled from $\mathcal{N}(0, Q)$; to generate $Q$, we randomly sample an $8 \times 8$ orthonormal matrix $U_Q$ and form $Q = U_Q \Sigma_Q U_Q^T$, where $\Sigma_Q$ is a diagonal matrix whose entries are sampled from the uniform distribution $\mathcal{U}[0, \sigma_q^2]$. For training, we implement a curriculum where $\sigma_q^2$ is steadily incremented over 100000 steps until reaching 0.025 and kept constant from then on. Similarly, the measurement noise is sampled from $\mathcal{N}(0, R)$ where the diagonal matrix $R$ has entries $\sigma_1^2, ..., \sigma_m^2$ sampled from $\mathcal{U}[0, \sigma_r^2]$. As in the case of the process noise, we steadily increase $\sigma_r^2$ over 100000 training steps until reaching 0.025. Note that $Q$ and $R$ are sampled anew for each example, To randomly generate the state matrix $F$, we explore two strategies:

1. **Strategy 1:** For the first set of experiments we set $F = (1 - \alpha)I + \alpha U_F$, where $\alpha \in \mathcal{U}[0, 1]$ and $U_F$ denotes a random orthonormal matrix. Note that the eigenvalues of matrix $F$ are in general complex valued and can thus be expressed as $pe^{j\phi}$, where $\phi$ denotes the phase of the said eigenvalue. As $\alpha$ increases from 0 to 1, we observe $\phi$ varying from 0 to $\pi$; here $\phi$ is such that the phase of the eigenvalues of $F$ is distributed in the interval $[-\phi, \phi]$. For $\alpha = 0$ and $\alpha = 1$, all the eigenvalues lie exactly on the unit circle; for the values of $\alpha \in (0, 1)$, the eigenvalues may lie inside the unit circle. Note that the dynamical system is not guaranteed to be stable since the eigenvalues of $F$ may lie on the unit circle. In fact, we observe that if one sets $\alpha = 1$, the transformer's loss does not decrease. To train the transformer, we steadily increase $\alpha$ from 0 to 1 over 50000 steps and then keep it constant.

2. **Strategy 2:** We further explore the setting with $F = U_F \Sigma_F U_F^T$, where $U_F$ denotes a random orthonormal matrix and the diagonal matrix $\Sigma_F$ has its entries drawn from $\mathcal{U}[-1, 1]$. The dynamical system with state matrix $F$ defined this way is guaranteed to be stable.

To compare the transformer's performance with that of other algorithms, we utilize the mean-squared prediction difference (MSPD) which relates the performance of two algorithms $\mathcal{A}_1$ and $\mathcal{A}_2$ given the same context $\mathcal{D}$ as

$$MSPD(\mathcal{A}_1, \mathcal{A}_2) = \mathbb{E}_{\mathcal{D}=[H_1, \ldots, H_{N-1}] \sim p(\mathcal{D}), h_N \sim p(h)} (\mathcal{A}_1(\mathcal{D})(H_N) - \mathcal{A}_2(\mathcal{D})(H_N))^2. \quad (29)$$

For evaluation, we utilize a randomly sampled batch of 5000 examples. Unless stated otherwise, the parameters of the state and measurement noise covariance are set to $\sigma_q^2 = \sigma_r^2 = 0.025$. Finally, for the Kalman filter used as the baseline, we set the estimate of the initial state to zero and the corresponding error covariance matrix to identity.

Before we delve into in-context learning, a natural question to ask would be whether the transformers can perform explicit state estimation given the input in the form of expression (67). To this end, we train the transformer to output the state for scalar measurements and Strategy 1 using the loss function

$$\frac{1}{N} \sum_{t=1}^{N} (x_t - T_\theta(h_1, y_1, ..., h_{t-1}, y_{t-1}, h_t, F, Q, \sigma^2))^2. \tag{30}$$

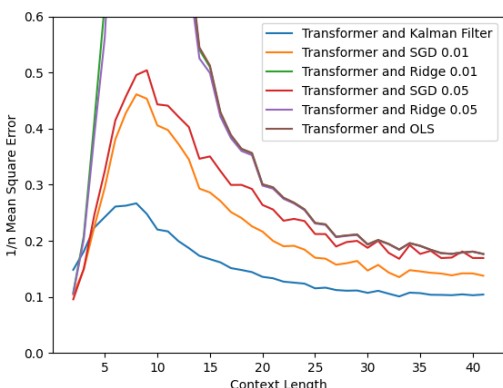

Figure 1: *The results of experiments testing the performance of various methods on the task of estimating states of a linear dynamical system. The plots compare the difference in the achieved mean-squared error between the transformer and other methods.*

The mean-squared error between the estimate of $x_N$ returned by the transformer and other algorithms is shown in Figure 6. The algorithms include the Kalman filter, stochastic gradient descent with learning rates of 0.01 and 0.05, ridge regression with regularization parameter $\lambda$ set to 0.01 and 0.05, and Ordinary Least Squares. A discussion addressing the choice of learning rates is provided in the appendix. It can be observed that as the context length increases, performance of the transformer approaches that of Kalman filter while diverging away from the Ridge Regression and the OLS; this is unsurprising since the latter two methods focus only on the measurement equation while remaining unaware of the internal state dynamics. In fact, when the context length equals the state dimension, the MSPD between the transformer and ridge regression is an order of magnitude higher than that between the transformer and Kalman filter; for better visualization/resolution, we thus limit the values on the vertical axis of this and other presented figures (leading to clipping in some of the plots).

Next, we focus on the problem of in-context learning where the model must make a one-step prediction of the system output given the context. The results are presented in Figure 2. For shorter contexts, the in-context learning performs closest to the stochastic gradient descent with learning rate 0.01, but as the context length increases, the performance of in-context learning approaches that of the Kalman filter. Note that when the stability is guaranteed, i.e., all the eigenvalues of $F$ are between 0 and 1 (Strategy 2), the performance gaps are smaller than when the stability may be violated.

To test the robustness of the transformers in face of partially missing context, we investigate what happens if the covariance matrices $R$ and $Q$ are omitted from the context and repeat the previous experiments. The results are reported in Figure 3. Interestingly, there appears to be no deterioration in the performance for Strategy 1 (see Figure 3a), and an improvement in the MSPD between the transformer performing in-context learning (ICL) and Kalman filter for Strategy 2 (Figure 3b). This may be implying that the transformer implicitly learns the missing context en route to mimicking

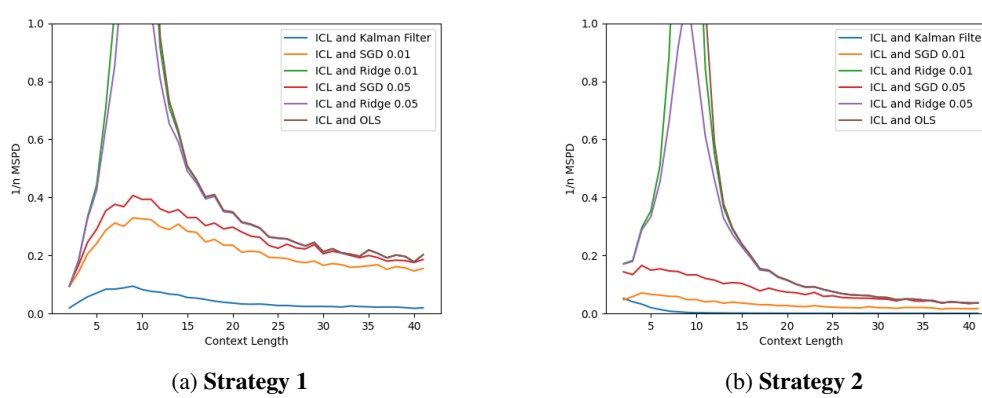

(a) **Strategy 1**  (b) **Strategy 2**

Figure 2: *Mean-squared prediction difference (MSPD) between in-context learning (ICL) with a transformer and several algorithms including Kalman filter, SGD, Ridge Regression, and OLS (scalar measurements).*

Kalman filter. Note that in these experiments the Kalman filter is still provided information about the noise statistics. If it were not, one would need to employ a technique such as the computationally intensive expectation-maximization algorithm to infer the missing noise covariance matrices.

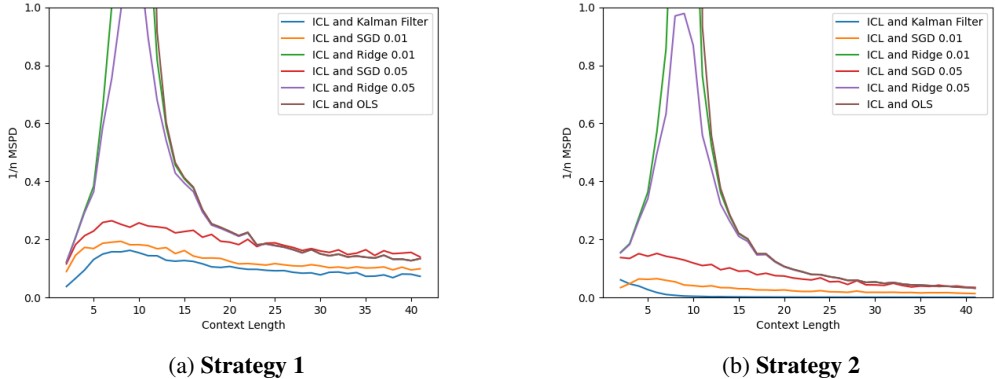

(a) **Strategy 1**  (b) **Strategy 2**

Figure 3: *Mean-squared prediction difference (MSPD) between in-context learning (ICL) with a transformer and several algorithms including Kalman filter, SGD, Ridge Regression, and OLS (scalar measurements). ICL is conducted without information about the covariances R and Q.*

We next investigate in-context learning for non-scalar measurements (dimension = 2) with white noise. The input to the transformer, formatted according to expression (28), includes all the parameters of the state space model. The results, presented in Figure 4, show that transformer is able to mimic Kalman filter following in-context learning and performs one-step prediction in the considered non-scalar measurements case.

Our final set of experiments investigates the setting where we withhold from the transformer the information about both the state transition matrix and noise covariances (for simplicity, we are back to the scalar measurements case). The results, plotted in Figure 5, show that the transformer starts emulating the Kalman filter for sufficiently large context lengths. In the appendix we provide theoretical arguments that in the case of scalar measurements, the transformer is able to implement operations of the Dual-Kalman Filter (Wan & Nelson, 1996), which allows implicit estimation of both the state and state transition matrix.

It is relatively straightforward to extend the presented results to the systems with control inputs and to certain classes of non-linear systems; for the latter, it can be shown that the transformer can implement operations of the Extended Kalman Filter (EKF). Relevant experimental results are

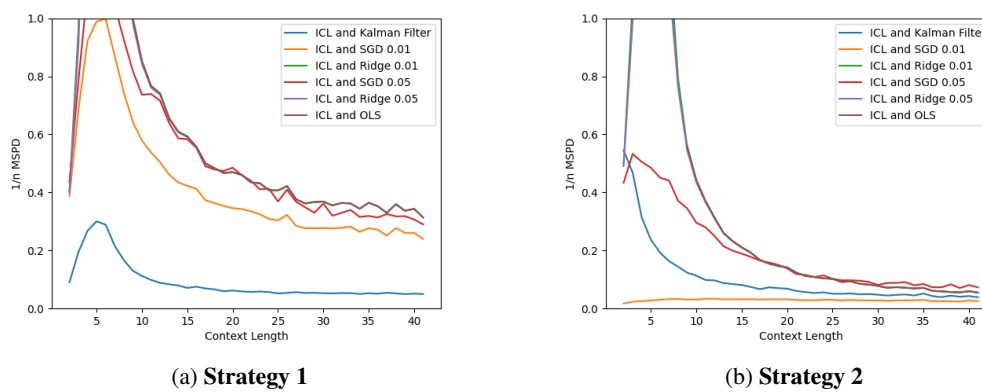

(a) **Strategy 1**    (b) **Strategy 2**

Figure 4: *Mean-squared prediction difference (MSPD) between in-context learning (ICL) with a transformer and several algorithms including Kalman filter, SGD, Ridge Regression, and OLS with the measurements of dimension 2. Note that for these experiments, all system parameters are available to the transformer.*

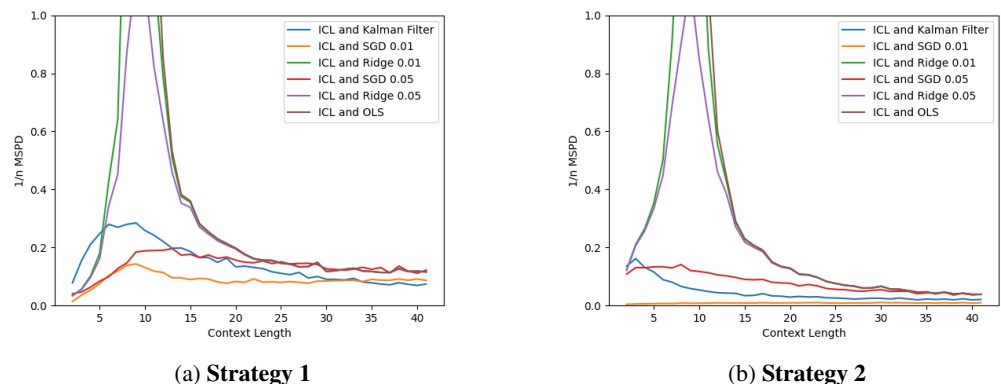

(a) **Strategy 1**    (b) **Strategy 2**

Figure 5: *Mean-squared prediction difference (MSPD) between in-context learning (ICL) with a transformer and several algorithms including Kalman filter, SGD, Ridge Regression, and OLS. All the information about the model parameters is withheld from the transformer.*

presented in the appendix. There, we also demonstrate robustness of in-context learning to variations in state dimensions or model parameter distributions as compared to those seen during training.

## 5    CONCLUSION

In this paper, we explored the capability of transformers to emulate the behavior of Kalman filter when trained in-context with the randomly sampled parameters of a state space model and the corresponding observations. We provided analytical arguments in support of the transformer's ability to do so, and presented empirical results of the experiments that demonstrate close proximity of the transformer and Kalman filter when the transformer is given sufficiently long context. Notably, the transformer keeps closely approximating Kalman filter even when important context – namely, noise covariance matrices and even state transition matrix (all required by the Kalman filter) – is omitted, demonstrating robustness and implying the ability to implicitly learn missing context. Future work includes extensions to temporally correlated noise and further investigation of robustness to missing model parameters.

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

## A   DUAL KALMAN FILTER - REVIEW

Consider the state-space model

$$x_{t+1} = F_t x_t + q_t \tag{31}$$
$$y_t = H_t x_t + r_t, \tag{32}$$

and consider the setting where we need to estimate both the state $x_t$ and the state transition matrix $F_t$. A solution is given in the form of the Dual Kalman Filter, proposed by Wan & Nelson (1996). This approach alternates between estimation of the state given the estimate of the state transition matrix and vice versa. Let $f_t \in \mathbb{R}^{n^2}$ denote the vectorized form of the state transition matrix, and let the $n \times n^2$ matrix $X_t$ be defined as

$$X_t = \begin{bmatrix} \hat{x}_{t-1}^{+T} & 0 & \dots & 0 & 0 \\ 0 & \hat{x}_{t-1}^{+T} & \dots & 0 & 0 \\ . & . & \dots & . & . \\ . & . & \dots & . & . \\ 0 & 0 & \dots & \hat{x}_{t-1}^{+T} & 0 \\ 0 & 0 & \dots & 0 & \hat{x}_{t-1}^{+T} \end{bmatrix}. \tag{33}$$

Then the following state space model can be set up for $f_t$:

$$f_t = f_{t-1} \tag{34}$$
$$y_t = H_{f,t} f_{t-1} + r_{f,t}, \tag{35}$$

where $H_{f,t} = H_t X_t$ and $r_{f,t} = H_t q_t + r_t$; moreover, $r_{f,t} \sim \mathcal{N}(0, R_f)$ where $R_f = H_t Q H_t^T + R$. The prediction and update equations for the estimates of $f_t$ are as follows.

**Prediction Step**:

$$\hat{f}_t^- = \hat{f}_{t-1}^+ \tag{36}$$
$$\hat{P}_{f,t}^- = \hat{P}_{f,t-1}^+ \tag{37}$$

**Update Step**:

$$K_{f,t} = \hat{P}_{f,t}^- H_{t,f}^T (H_{f,t} \hat{P}_{f,t}^- H_{t,f}^T + R_f)^{-1} \tag{38}$$
$$\hat{f}_t^+ = \hat{f}_t^- + K_{f,t}(y_t - H_{f,t} \hat{f}_t^-) \tag{39}$$
$$\hat{P}_{f,t}^+ = (I - K_{f,t} H_{f,t}) \hat{P}_{f,t}^- \tag{40}$$

In case of scalar measurements, $H_{f,t} \hat{P}_{f,t}^- H_{t,f}^T$, and $H_t Q H_t^T$ and $R$ are scalar; consequently, we can express (38) as

$$K_{f,t} = \frac{1}{H_{f,t} \hat{P}_{f,t}^- H_{t,f}^T + R_f} \hat{P}_{f,t}^- H_{t,f}^T. \tag{41}$$

## B   TRANSFORMER CAN LEARN TO PERFORM DUAL KALMAN FILTERING IN-CONTEXT FOR A SYSTEM WITH SCALAR MEASUREMENTS

Consider the following context provided to the transformer as input:

$$\begin{bmatrix} 0 & \sigma^2 & 0 & y_1 & 0 & y_2 & \dots & y_{N-1} & 0 \\ Q & 0 & h_1^T & 0 & h_2^T & 0 & \dots & 0 & h_N^T \end{bmatrix}. \tag{42}$$

One can show in a manner analogous to the proof for the canonical Kalman filter that the transformer can in-context learn to perform implicit state estimation even in the absence of the state transition matrix. To this end, in addition to **Mul**(I, J, K). **Div**(I, j, K), **Aff**(I, J, K, W1, W2), and **Transpose**(I, J) defined in the main text, we define **MAP**(I,J) which transforms the vectors formed by the entries at index set $I$ to a matrix of the form in equation (33) to be copied and stored at the index set $J$.

In addition to $\mathcal{A}_{cat}, I_F, I_Q, I_\sigma, I_{B1}, I_{B2}, I_{B3}, I_{B4}, I_{B5}, I_{B6}, I_{B7}, I_{B8}, I_{B9}$ defined in the main text, we let $I_{B10}$ denote indices of an $n \times n^2$ sub-matrix of zeros in $\mathcal{A}_{cat}$; let $I_{B11}$ denote indices of a $1 \times n^2$ sub-matrix of zeros in $\mathcal{A}_{cat}$; $I_{B13}$ and $I_{\hat{f}_{next}}$ denote indices of an $n^2 \times 1$ sub-matrices of zeros; and $I_{B12}$ and $I_{B14}$ denotes indices of an $n^2 \times n^2$ sub-matrices of zeros. With this notation, we can re-write the Dual Kalman filter using the elementary operations implementable by transformers as Algorithm 2.

---

**Algorithm 2** Formulating the Dual Kalman filter recursions using the elementary operations implementable by transformers

---

1: **Input**: $\mathcal{A}_{cat}, I_F, I_Q, I_\sigma, I_{B1}, I_{B2}, I_{B3}, I_{B4}, I_{B5}, I_{B6}, I_{B7}, I_{B8}, I_{B9}, I_{B10}, I_{B11}, I_{B12}, I_{B13}, I_{B14}, I_{\hat{f}_{next}}$

2: **Initialize** $I_{\hat{X}_{Curr}} \leftarrow (1 : n, 2n)$

3: **for** $i = 1$ to $N$ **do**

4:      $I_{\hat{X}_{next}} \leftarrow (1 : n, 2n + 2i)$

5:      $I_h \leftarrow (1 : n, 2n + 2i - 1)$

6:      $I_y \leftarrow (0, 2n + 2i)$

7:      **Transpose**($I_F, I_{B2}$)

8:      **Mul**($I_F, I_{\hat{X}_{Curr}}, I_{\hat{X}_{next}}$)

9:      **Mul**($I_F, I_{B1}, I_{B1}$)

10:      **Mul**($I_{B1}, I_{B2}, I_{B1}$)

11:      **Aff**($I_{B1}, I_Q, I_{B1}, W_1 = I_{n \times n}, W_2 = I_{n \times n}$)

12:      **Transpose**($I_h, I_{B3}$)

13:      **Mul**($I_{B1}, I_h, I_{B4}$)

14:      **Mul**($I_{B3}, I_{B4}, I_{B5}$)

15:      **Aff**($I_{B5}, I_\sigma, I_{B6}, W_1 = 1, W_2 = 1$)

16:      **Div**($I_{B4}, I_{B6}, I_{B4}$)

17:      **Mul**($I_h, I_{\hat{X}_{next}}, I_{B7}$)

18:      **Aff**($I_y, I_{B7}, I_{B7}, W_1 = 1, W_2 = -1$)

19:      **Mul**($I_{B7}, I_{B4}, I_{B8}$)

20:      **Aff**($I_{\hat{X}_{next}}, I_{B8}, I_{\hat{X}_{next}}, W_1 = 1, W_2 = 1$)

21:      **Mul**($I_{B4}, I_{B3}, I_{B9}$)

22:      **Mul**($I_{B9}, I_{B1}, I_{B9}$)

23:      **Aff**($I_{B1}, I_{B9}, I_{B1}, W_1 = I_{n \times n}, W_2 = -I_{n \times n}$)

24:      **MAP**($I_{\hat{X}_{next}}, I_{B10}$)

25:      **Mul**($I_{B3}, I_{B10}, I_{B11}$)

26:      **Transpose**($I_{B11}, I_{B13}$)

27:      **Mul**($I_{B11}, I_{B12}, I_{B11}$)

28:      **Mul**($I_{B11}, I_{B13}, I_{B5}$)

29:      **Mul**($I_{B12}, I_{B13}, I_{B13}$)

30:      **Aff**($I_{B5}, I_\sigma, I_{B6}, W_1 = 1, W_2 = 1$)

31:      **Div**($I_{B13}, I_{B6}, I_{B13}$)

32:      **Mul**($I_{B7}, I_{B13}, I_{B8}$)

33:      **Aff**($I_{\hat{f}_{next}}, I_{B8}, I_{\hat{f}_{next}}, W_1 = 1, W_2 = 1$)

34:      **Mul**($I_{B3}, I_{B10}, I_{B11}$)

35:      **Mul**($I_{B13}, I_{B11}, I_{B14}$)

36:      **Mul**($I_{B14}, I_{B12}, I_{B14}$)

37:      **Aff**($I_{B12}, I_{B14}, I_{B12}, W_1 = I_{n^2 \times n^2}, W_2 = -I_{n^2 \times n^2}$)

38:      **MAP**($I_{\hat{f}_{next}}, I_F$)

39:      $I_{\hat{X}_{Curr}} \leftarrow I_{\hat{X}_{next}}$

40: **end for**

---

## C    EXPERIMENTS WITH CONTROL INPUT

The results presented in the main body of the paper can be extended to the case of non-zero control inputs provided that the measurement noise remains white. Specifically, for non-zero control inputs, the state space model becomes

$$x_{t+1} = F_t x_t + B_t u_t + q_t \tag{43}$$
$$y_t = H_t x_t + r_t. \tag{44}$$

The derivation showing that a transformer can implement operations of Kalman filtering for this state space model follows the same line of arguments presented in the main body of the paper for other (simpler) state space models. We omit details for brevity and instead focus on presenting empirical results. In particular, we carry out experiments involving scalar measurements and generate $B \in \mathbb{R}^{8 \times 8}$ as $B = U_B \Sigma_B U_B^T$, where $U_B$ denotes a random orthonormal matrix while the diagonal matrix $\Sigma_B$ has entries drawn from $\mathcal{U}[-1, 1]$. We sample control inputs $u_t \in \mathbb{R}^8$ from a zero-mean Gaussian distribution having identity matrix as the covariance, and then normalize them to the unit norm. The input to the transformer for this setting is

$$\begin{matrix} 0 & 0 & 0 & \sigma^2 & 0 & 0 & 0 & y_1 & \cdots & 0 & 0 \\ F & Q & B & 0 & 0 & h_1^T & u_1 & 0 & \cdots & h_N^T & u_N \end{matrix} \tag{45}$$

where F is generated using strategy 1. The rest of the settings remain the same as in the previous experiments. The results, reported in Fig. 6, demonstrate that the transformer achieves mean-square error performance similar to that of Kalman filter in this setting as well.

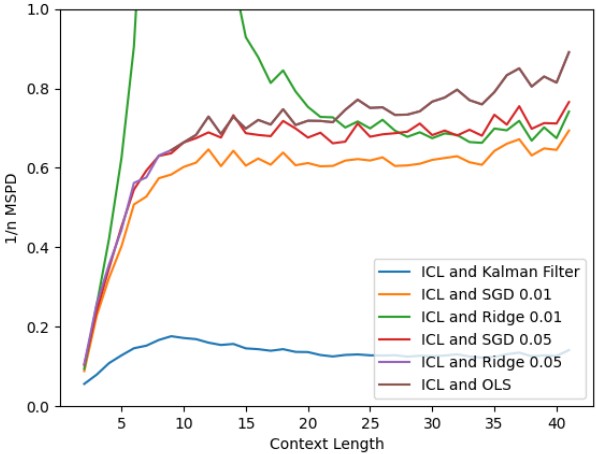

Figure 6: *The results of experiments with control inputs.*

## D    MISCELLANEOUS EXPERIMENTS AND FURTHER DETAILS

### D.1    DETAILED ILLUSTRATION OF ALGORITHM 1

We illustrate the working of Algorithm 1 for scalar measurements. Let the state dimension be $n$ and let the measurements be scalar. Let $\mathcal{A}_{append} =$

$$\begin{matrix} B_1 & B_2 & B_9 & B_3^T & B_4 & B_8 & 0_{n \times 1} & 0_{n \times 1} & 0_{n \times 1} \\ 0_{1 \times n} & 0_{1 \times n} & 0_{1 \times n} & 0 & 0 & 0 & B_5 & B_6 & B_7 \end{matrix} \tag{46}$$

For $n = 2$, this can be visualized as $\mathcal{A}_{append} =$

$$\begin{matrix} 1 & 0 & 0 & 0 & 0 & 0 & 0 & 0 & 0 & 0 & 0 & 0 \\ 0 & 1 & 0 & 0 & 0 & 0 & 0 & 0 & 0 & 0 & 0 & 0 \\ 0 & 0 & 0 & 0 & 0 & 0 & 0 & 0 & 0 & 0 & 0 & 0 \end{matrix} \tag{47}$$

Consequently, with $\mathcal{A}_{cat} = [\mathcal{A}_{append}\ \mathcal{A}_{input}]$, we obtain $I_{B1} = \{(0,0),(1,0),(0,1),(1,1)\}$, $I_{B2} = \{(0,2),(1,2),(0,3),(1,3)\}$, $I_{B9} = \{(0,4),(1,4),(0,5),(1,5)\}$, $I_{B3} = \{(0,6),(1,6)\}$, $I_{B4} = \{(0,7),(1,7)\}$, $I_{B9} = \{(0,8),(1,8)\}$, $I_{B5} = \{(2,9)\}$, $I_{B6} = \{(2,10)\}$, $I_{B7} = \{(2,11)\}$, $I_F = \{(1,12),(1,13),(2,12),(2,13)\}$, $I_Q = \{(1,14),(1,15),(2,14),(2,15)\}$, $I_\sigma = \{(0,16)\}$ and so on.

Next, we walk through the first iteration of the FOR loop in Algorithm 1.

1. Initialization: $I_{\hat{X}_{Curr}} \leftarrow (1:n, 2n)$. We start by initializing $\hat{x}_0^+ = 0$. We simply use the zeros below the variances as denoted below

$$
\begin{bmatrix}
0 & 0 & \sigma^2 & 0 & y_1 & 0 & y_2 & ... & y_{N-1} & 0 \\
F & Q & \hat{x}_{Curr}=0 & h_1^T & 0 & h_2^T & 0 & ... & 0 & h_N^T
\end{bmatrix}.
\tag{48}
$$

2. $I_{\hat{X}_{next}} \leftarrow (1:n, 2n+2i)$. For the first iteration, $i=1$, this points to the elements just below the first measurement

$$
\begin{bmatrix}
0 & 0 & \sigma^2 & 0 & y_1 & 0 & y_2 & ... & y_{N-1} & 0 \\
F & Q & \hat{x}_{Curr}=0 & h_1^T & \hat{x}_{next}=0 & h_2^T & 0 & ... & 0 & h_N^T
\end{bmatrix}.
\tag{49}
$$

3. $I_h \leftarrow (1:n, 2n+2i-1)$ This points to $h_1^T$ in $\mathcal{A}_{input}$ Likewise, $I_y \leftarrow (0, 2n+2i)$ points to $y_1$.

4. **Transpose($I_F, I_{B2}$)** This writes $F$ to matrix $B_2$. The matrix $\mathcal{A}_{append}$ becomes

$$
\begin{matrix}
B_1 & F^T & B_9 & B_3^T & B_4 & B_8 & 0_{n\times 1} & 0_{n\times 1} & 0_{n\times 1} \\
0_{1\times n} & 0_{1\times n} & 0_{1\times n} & 0 & 0 & 0 & B_5 & B_6 & B_7
\end{matrix}
\tag{50}
$$

5. **Mul($I_F, I_{\hat{X}_{Curr}}, I_{\hat{X}_{next}}$)**. This calculates $F\hat{x}_{Curr}$ and writes to $I_{\hat{X}_{next}}$. $\mathcal{A}_{input}$ becomes

$$
\begin{bmatrix}
0 & 0 & \sigma^2 & 0 & y_1 & 0 & y_2 & ... & y_{N-1} & 0 \\
F & Q & \hat{x}_{Curr}=0 & h_1^T & F\hat{x}_{Curr}=0 & h_2^T & 0 & ... & 0 & h_N^T
\end{bmatrix}.
\tag{51}
$$

6. **Mul($I_F, I_{B1}, I_{B1}$)**. For the first iteration, $B1 = \hat{P}_0^+ = I$ This operation calculates $F\hat{P}_0^+$ and writes it to $B_1$. The resulting $\mathcal{A}_{append}$ becomes

$$
\begin{matrix}
F\hat{P}_0^+ & F^T & B_9 & B_3^T & B_4 & B_8 & 0_{n\times 1} & 0_{n\times 1} & 0_{n\times 1} \\
0_{1\times n} & 0_{1\times n} & 0_{1\times n} & 0 & 0 & 0 & B_5 & B_6 & B_7
\end{matrix}
\tag{52}
$$

7. **Mul($I_{B1}, I_{B2}, I_{B1}$)**. This calculates $F\hat{P}_0^+ F^T$ and writes to $I_{B1}$.

$$
\begin{matrix}
F\hat{P}_0^+ F^T & F^T & B_9 & B_3^T & B_4 & B_8 & 0_{n\times 1} & 0_{n\times 1} & 0_{n\times 1} \\
0_{1\times n} & 0_{1\times n} & 0_{1\times n} & 0 & 0 & 0 & B_5 & B_6 & B_7
\end{matrix}
\tag{53}
$$

8. **Aff($I_{B1}, I_Q, I_{B1}, W_1 = I_{n\times n}, W_2 = I_{n\times n}$)**. This calculates $\hat{P}_1^- = F\hat{P}_0^+ F^T + Q$ and writes to $I_{B1}$

$$
\begin{matrix}
\hat{P}_1^- & F^T & B_9 & B_3^T & B_4 & B_8 & 0_{n\times 1} & 0_{n\times 1} & 0_{n\times 1} \\
0_{1\times n} & 0_{1\times n} & 0_{1\times n} & 0 & 0 & 0 & B_5 & B_6 & B_7
\end{matrix}
\tag{54}
$$

9. **Transpose($I_h, I_{B3}$)** This transposes $h_1^T$ and writes to $B_3$ which yields

$$
\begin{matrix}
\hat{P}_1^- & F^T & B_9 & h_1^T & B_4 & B_8 & 0_{n\times 1} & 0_{n\times 1} & 0_{n\times 1} \\
0_{1\times n} & 0_{1\times n} & 0_{1\times n} & 0 & 0 & 0 & B_5 & B_6 & B_7
\end{matrix}
\tag{55}
$$

10. **Mul($I_{B1}, I_h, I_{B4}$)**. This evaluates $\hat{P}_1^- h_1^T$ and writes it to $I_{B4}$ which yields

$$
\begin{matrix}
\hat{P}_1^- & F^T & B_9 & h_1^T & \hat{P}_1^- h_1^T & B_8 & 0_{n\times 1} & 0_{n\times 1} & 0_{n\times 1} \\
0_{1\times n} & 0_{1\times n} & 0_{1\times n} & 0 & 0 & 0 & B_5 & B_6 & B_7
\end{matrix}
\tag{56}
$$

11. **Mul**($I_{B3}, I_{B4}, I_{B5}$). This evaluates scalar $h_1 \hat{P}_1^- h_1^T$ and writes to $I_{B5}$ yielding

$$
\begin{matrix}
\hat{P}_1^- & F^T & B_9 & h_1^T & \hat{P}_1^- h_1^T & B_8 & 0_{n\times 1} & 0_{n\times 1} & 0_{n\times 1} \\
0_{1\times n} & 0_{1\times n} & 0_{1\times n} & 0 & 0 & 0 & h_1 \hat{P}_1^- h_1^T & B_6 & B_7
\end{matrix}
\tag{57}
$$

12. **Aff**($I_{B5}, I_\sigma, I_{B6}, W_1 = 1, W_2 = 1$) This evaluates $h_1 \hat{P}_1^- h_1^T + \sigma^2$ and writes the resulting scalar to $I_{B6}$ resulting in

$$
\begin{matrix}
\hat{P}_1^- & F^T & B_9 & h_1^T & \hat{P}_1^- h_1^T & B_8 & 0_{n\times 1} & 0_{n\times 1} & 0_{n\times 1} \\
0_{1\times n} & 0_{1\times n} & 0_{1\times n} & 0 & 0 & 0 & h_1 \hat{P}_1^- h_1^T & h_1 \hat{P}_1^- h_1^T + \sigma^2 & B_7
\end{matrix}
\tag{58}
$$

13. **Div**($I_{B4}, I_{B6}, I_{B4}$) This divides the entries in $\hat{P}_1^- h_1^T$ by the scalar $h_1 \hat{P}_1^- h_1^T + \sigma^2$ to compute the Kalman Gain $K_1$ and writes the results to $I_{B4}$ giving

$$
\begin{matrix}
\hat{P}_1^- & F^T & B_9 & h_1^T & K_1 = \frac{1}{h_1 \hat{P}_1^- h_1^T + \sigma^2} \hat{P}_1^- h_1^T & B_8 & 0_{n\times 1} & 0_{n\times 1} & 0_{n\times 1} \\
0_{1\times n} & 0_{1\times n} & 0_{1\times n} & 0 & 0 & 0 & h_1 \hat{P}_1^- h_1^T & h_1 \hat{P}_1^- h_1^T + \sigma^2 & B_7
\end{matrix}
\tag{59}
$$

14. **Mul**($I_h, I_{\hat{X}_{next}}, I_{B7}$) This evaluate $h_1 \hat{x}_1^- = h_1 F \hat{x}^+ 0 = h_1 F \hat{x}_{Curr}$ and writes the resulting scalar to $I_{B7}$.

$$
\begin{matrix}
\hat{P}_1^- & F^T & B_9 & h_1^T & K_1 & B_8 & 0_{n\times 1} & 0_{n\times 1} & 0_{n\times 1} \\
0_{1\times n} & 0_{1\times n} & 0_{1\times n} & 0 & 0 & 0 & h_1 \hat{P}_1^- h_1^T & h_1 \hat{P}_1^- h_1^T + \sigma^2 & h_1 \hat{x}_1^-
\end{matrix}
\tag{60}
$$

15. **Aff**($I_y, I_{B7}, I_{B7}, W_1 = 1, W_2 = -1$) This evaluates $y_1 - h_1 \hat{x}_1^-$ and writes it to $I_{B7}$

$$
\begin{matrix}
\hat{P}_1^- & F^T & B_9 & h_1^T & K_1 & B_8 & 0_{n\times 1} & 0_{n\times 1} & 0_{n\times 1} \\
0_{1\times n} & 0_{1\times n} & 0_{1\times n} & 0 & 0 & 0 & h_1 \hat{P}_1^- h_1^T & h_1 \hat{P}_1^- h_1^T + \sigma^2 & y_1 - h_1 \hat{x}_1^-
\end{matrix}
\tag{61}
$$

16. **Mul**($I_{B7}, I_{B4}, I_{B8}$) This multiplies the Kalman gain with the error to yield

$$
\begin{matrix}
\hat{P}_1^- & F^T & B_9 & h_1^T & K_1 & K_1(y_1 - h_1 \hat{x}_1^-) & 0_{n\times 1} & 0_{n\times 1} & 0_{n\times 1} \\
0_{1\times n} & 0_{1\times n} & 0_{1\times n} & 0 & 0 & 0 & h_1 \hat{P}_1^- h_1^T & h_1 \hat{P}_1^- h_1^T + \sigma^2 & y_1 - h_1 \hat{x}_1^-
\end{matrix}
\tag{62}
$$

17. **Aff**($I_{\hat{X}_{next}}, I_{B8}, I_{\hat{X}_{next}}, W_1 = 1, W_2 = 1$) This calculates the posterior estimate $\hat{x}_1^+ = \hat{x}_1^- + K_1(y_1 - h_1 \hat{x}_1^-)$ and writes it to $I_{\hat{X}_{next}}$ modifying $\mathcal{A}_{input}$ to

$$
\begin{bmatrix}
0 & 0 & \sigma^2 & 0 & y_1 & 0 & y_2 & ... & y_{N-1} & 0 \\
F & Q & \hat{x}_{Curr} = 0 & h_1^T & \hat{x}_1^+ & h_2^T & 0 & ... & 0 & h_N^T
\end{bmatrix}.
\tag{63}
$$

18. **Mul**($I_{B4}, I_{B3}, I_{B9}$) This evaluates the $n \times n$ matrix $h_1 K_1$ and writes it to $I_{B9}$ giving us

$$
\begin{matrix}
\hat{P}_1^- & F^T & h_1 K_1 & h_1^T & K_1 & K_1(y_1 - h_1 \hat{x}_1^-) & 0_{n\times 1} & 0_{n\times 1} & 0_{n\times 1} \\
0_{1\times n} & 0_{1\times n} & 0_{1\times n} & 0 & 0 & 0 & h_1 \hat{P}_1^- h_1^T & h_1 \hat{P}_1^- h_1^T + \sigma^2 & y_1 - h_1 \hat{x}_1^-
\end{matrix}
\tag{64}
$$

19. **Mul**($I_{B9}, I_{B1}, I_{B9}$): This calculates and writes to $I_{B9}$ the matrix $h_1 K_1 \hat{P}_1^-$

$$
\begin{matrix}
\hat{P}_1^- & F^T & h_1 K_1 \hat{P}_1^- & h_1^T & K_1 & K_1(y_1 - h_1 \hat{x}_1^-) & 0_{n\times 1} & 0_{n\times 1} & 0_{n\times 1} \\
0_{1\times n} & 0_{1\times n} & 0_{1\times n} & 0 & 0 & 0 & h_1 \hat{P}_1^- h_1^T & h_1 \hat{P}_1^- h_1^T + \sigma^2 & y_1 - h_1 \hat{x}_1^-
\end{matrix}
\tag{65}
$$

20. **Aff**($I_{B1}, I_{B9}, I_{B1}, W_1 = I_{n\times n}, W_2 = -I_{n\times n}$) This calculates the error covariance of the posterior estimate $\hat{P}_1^+ = \hat{P}_1^- - h_1 K_1 \hat{P}_1^-$ and writes the results to $I_{B9}$

$$
\begin{matrix}
\hat{P}_1^- & F^T & \hat{P}_1^+ & h_1^T & K_1 & K_1(y_1 - h_1 \hat{x}_1^-) & 0_{n\times 1} & 0_{n\times 1} & 0_{n\times 1} \\
0_{1\times n} & 0_{1\times n} & 0_{1\times n} & 0 & 0 & 0 & h_1 \hat{P}_1^- h_1^T & h_1 \hat{P}_1^- h_1^T + \sigma^2 & y_1 - h_1 \hat{x}_1^-
\end{matrix}
\tag{66}
$$

21. $I_{\hat{X}_{Curr}} \leftarrow I_{\hat{X}_{next}}$ This updates the pointer $I_{\hat{X}_{Curr}}$ to point towards indices $I_{\hat{X}_{next}}$ yielding

$$
\begin{bmatrix}
0 & 0 & \sigma^2 & 0 & y_1 & 0 & y_2 & ... & y_{N-1} & 0 \\
F & Q & 0 & h_1^T & \hat{x}_{Curr} = \hat{x}_1^+ & h_2^T & 0 & ... & 0 & h_N^T
\end{bmatrix}.
\tag{67}
$$

## D.2 JUSTIFICATION OF THE CHOICE OF LEARNING RATES FOR SGD

To find the optimal set of learning rates for the experiments, we fix context length to $40$ and the state dimension to $8$. We generate $F$ using Strategy 1. The rest of the simulation settings remain the same as before. We compare the MSPD between the transformer's output and the SGD for various values of $\alpha$ (learning rates). We present the results in Table 1

| $\alpha$ | 0.00001 | 0.00005 | 0.0001 | 0.0005 | 0.001 | 0.005 | 0.01 | 0.05 | 0.1 |
|------|---------|---------|--------|--------|-------|-------|------|------|-----|
| MSPD | 1.0087 | 0.9999 | 0.9753 | 0.8882 | 0.8789 | 0.4766 | **0.2997** | **0.2887** | 2.8923 |

Table 1: MSPD corresponding to different $\alpha$

As can be seen, the optimal MSPD is obtained for $\alpha = 0.01$ and $\alpha = 0.05$. Consequently, throughout this work we report the results for these two learning rates only.

## D.3 MEAN SQUARE ERROR (MSE) WITH RESPECT TO THE GROUND TRUTH

In the main section of the paper, we omit the MSE of the output of the transformer, Kalman filter, and other baseline algorithms evaluated with respect to the ground truth. Here we present the MSE, normalized by the state dimension, for the default simulation setting with $F$ generated using Strategy 1; note that the results for Strategy 2 and varied parameters differ very little from the presented ones. As can be seen in Fig. 7, the MSE curves closely follow those for the normalized MSPD presented in the main paper.

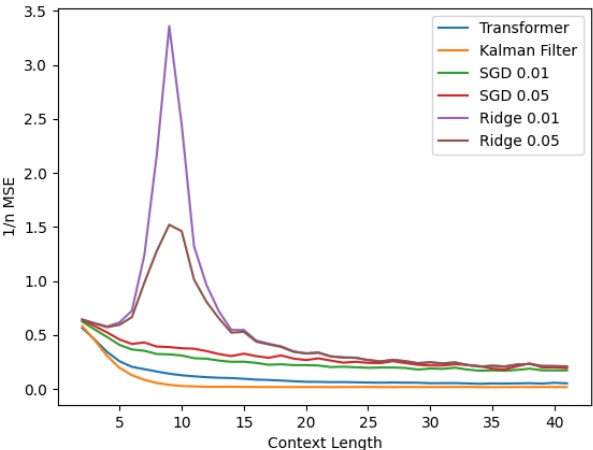

Figure 7: *The MSE between the output and the ground truth for various algorithms.*

## D.4 RESULTS ACROSS DIFFERENT STATE DIMENSIONS

To evaluate the performance of transformer as the state dimension varies, we train the transformer model under the default settings using Strategy 1 to generate F. For evaluation, we fix the context length to $40$ and vary state dimension from $2$ to $8$; we utilize Strategy 1 to generate F, while the remaining simulation parameters remain as same as before. The results in Fig. 8 present the MSPD normalized by the state dimension vs. varying state dimension. As can be seen there, the gap between the transformer and Kalman filter remains constant implying that the performance of the transformer remains consistent regardless of the state dimension. At the same time, the MSPD between the transformer and the SGD / Ridge Regression increases as the performances of the latter two algorithms deteriorate with increasing state dimension.

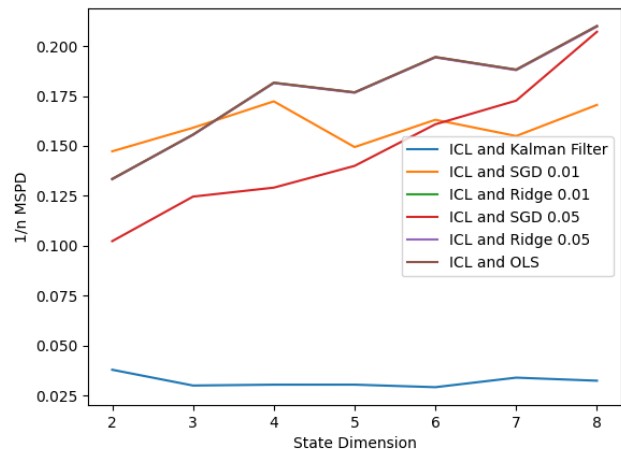

Figure 8: *Results for the fixed context length and varying state dimension*

### D.5 AN ILLUSTRATION OF THE PERFORMANCE ON OUT-OF-SAMPLE PARAMETERS

To evaluate the performance of the transformer on systems with parameters sampled from a distribution different from the one seen during the training, we train the transformer with $F$ generated using Strategy 1 as previously described but then evaluate the trained model in experiments conducted under following changes:

1. The measurement noise variance and the entries of $\Sigma_q$ are sampled from $\mathcal{U}[0.01, 0.05]$ instead of $\mathcal{U}[0, 0.025]$;

2. entries of $H_i$ are sampled from $\mathcal{U}[-\frac{5}{4}, \frac{5}{4}]$, $\mathcal{U}[0, 3]$ and $\mathcal{U}[1, 4]$ instead of $\mathcal{N}(0, 1)$;

3. the state transition matrix $F$ is generated via Strategy 2.

The results, presented in Fig. 9, demonstrate robustness of in-context learning to out-of-distribution parameters generated in the aforementioned way.

## E AN INVESTIGATION OF NON-LINEAR SYSTEMS

In this section, we consider systems with non-linear state space models of the form

$$x_{t+1} = f_\eta(x_t) + q_t \tag{68}$$
$$y_t = H_t x_t + r_t, \tag{69}$$

where $f_\eta()$ is a non-linear function parameterized by the set of parameters $\eta = [\eta_1, ..., \eta_w]$. To estimate the states, we can linearize the system and perform Extended Kalman Filtering.

**Prediction Step**:

$$\hat{x}_t^- = f_\eta(\hat{x}_{t-1}^+) \tag{70}$$
$$\hat{P}_t^- = \tilde{F}_x \hat{P}_{t-1}^+ \tilde{F}_x^T + Q \tag{71}$$

**Update Step**:

$$K_t = \hat{P}_t^- H_t^T (H_t \hat{P}_t^- H_t^T + R)^{-1} \tag{72}$$
$$\hat{x}_t^+ = \hat{x}_t^- + K_t(y_t - H_t \hat{x}_t^-) \tag{73}$$
$$\hat{P}_t^+ = (I - K_t H_t)\hat{P}_t^- \tag{74}$$

where $F_x$ is the Jacobian of $f_\eta()$ with respect to $\hat{x}_{t-1}^+$.

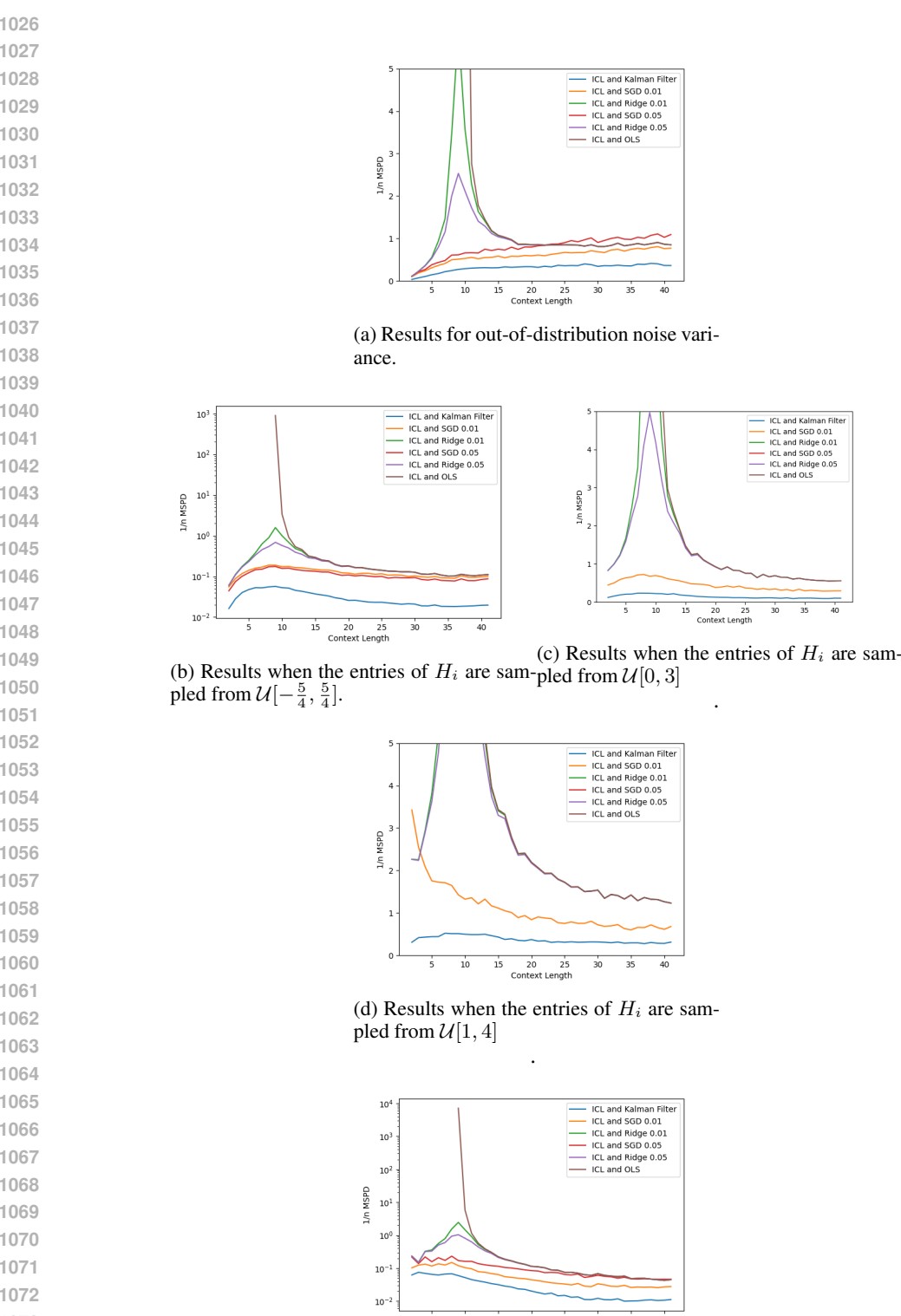

(a) Results for out-of-distribution noise variance.

(b) Results when the entries of $H_i$ are sampled from $\mathcal{U}[-\frac{5}{4}, \frac{5}{4}]$.

(c) Results when the entries of $H_i$ are sampled from $\mathcal{U}[0, 3]$

.

(d) Results when the entries of $H_i$ are sampled from $\mathcal{U}[1, 4]$

.

(e) Results for model trained under Strategy 1 evaluated on systems with $F$ generated using Strategy 2.

Figure 9: Evaluating trained transformer on systems with parameters drawn from distribution different from the one seen during the training.

For this work, we choose $\mathbf{f}_\eta(\mathbf{x}) = [\eta_1 \tanh(\eta_2 \mathbf{x_1}), ..., \eta_1 \tanh(\eta_2 \mathbf{x_n})]$ with $\eta_1, \eta_2 \sim \mathcal{U}[-1, 1]$. For this transition function, we can write the Jacobian as

$$\tilde{F}_x = \eta_1 \eta_2 diag([(1 - tanh^2(\eta_2 x_1)), ..., (1 - tanh^2(\eta_2 x_n))]).$$

For this system, we show that the transformer can theocratically implement equations of an Extended Kalman Filter given the input of the form

$$\begin{bmatrix} \eta_1 & \eta_2 & 0 & \sigma^2 & 0 & y_1 & ... & y_{N-1} & 0 \\ 0 & 0 & Q & 0 & h_1^T & 0 & ... & 0 & h_N^T \end{bmatrix}. \tag{75}$$

In Akyürek et al. (2023), the authors utilized the properties of GeLU non-linearity to show that GeLU can be used to perform scalar multiplication or for a sufficiently large additive bias term the identity function. Likewise, we can show in an analogous manner that

$$\frac{\sqrt{\frac{\pi}{2}}x}{2} tanh(x + cx^3) \approx GeLU(\sqrt{\frac{\pi}{2}}x) - GeLU(\frac{\sqrt{\frac{\pi}{2}}x}{2} + N_b) + N_b \tag{76}$$

where $N_b >> 1$ is a large bias term and the constant $c = \frac{0.044715\pi}{2}$. Since the softmax layer can be bypassed by adding a large additive bias term, it is straightforward to show that a single transformer attention head can output $\frac{\sqrt{\frac{\pi}{2}}x}{2} tanh(x + cx^3)$ for the input $x$ given appropriate parameters. Moreover, the simulation settings ensure that $x + cx^3 \approx x$, and consequently, raw operator's emulation of **Mul() Div()**, and **Aff()** operations can be invoked to argue that the transformer can implement $tanh(x)$ using multiple attention heads. It is then trivial to extend the method for the previously discussed linear dynamical systems case to show that the transformer can perform extended Kalman filtering for the non-linear dynamical system under consideration.

Given this setting, we run simulations with the parameters of the state and measurement noise co-variance set to $\sigma_q^2 = \sigma_r^2 = 0.0125$. We compare the performance of the transformer with that of the Extended Kalman Filter and present the results in Fig. 10. As can be seen there, the transformer performs closer to the Extended Kalman Filter than it does to other baseline algorithms. We leave investigation for other classes of non-linear systems to future work.

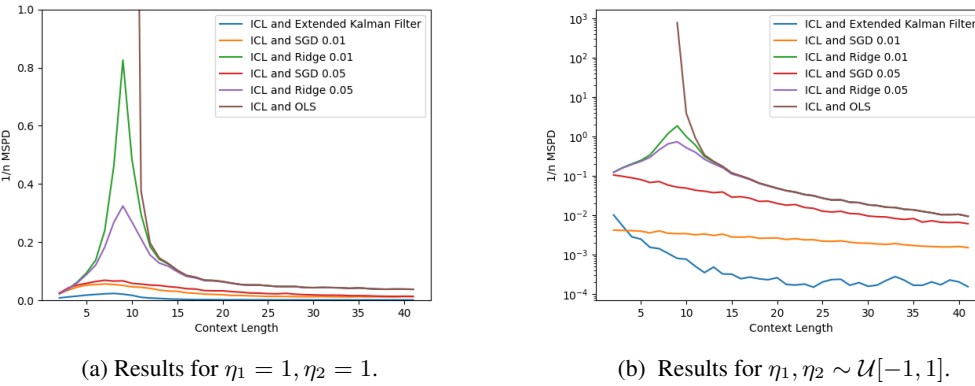

(a) Results for $\eta_1 = 1, \eta_2 = 1$.      (b) Results for $\eta_1, \eta_2 \sim \mathcal{U}[-1, 1]$.

Figure 10: *Results of in-context learning for $\mathbf{f}_\eta(\mathbf{x}) = [\eta_1 \tanh(\eta_2 \mathbf{x_1}), ..., \eta_1 \tanh(\eta_2 \mathbf{x_n})]$.*

