# OpenReview forum: "CAN TRANSFORMERS IN-CONTEXT LEARN BEHAVIOR OF A LINEAR DYNAMICAL SYSTEM?"
_ICLR.cc/2025/Conference — Submitted to ICLR 2025_

### Official Review · Reviewer_AMcU · 2024-10-28

**Soundness:** 2
**Presentation:** 2
**Contribution:** 2
**Rating:** 3
**Confidence:** 3

**Summary:**

The authors investigate the capability of transformer models to learn and mimic state estimation in linear dynamical systems (LDS). Building on Akyürek et al. (2022), who explored in-context learning for linear models, the authors demonstrate that transformers can effectively approximate the Kalman filter—the standard closed-form solution for linear dynamical systems. Through empirical analysis, they show that transformers successfully mimic Kalman filter behavior, providing accurate predictions for both latent states $(x_{1}, \dots, x_{n})$ and future observations $(y_{n+1})$. The study further explores how in-context learning (ICL) can replicate more challenging algorithms, like the Dual-Kalman filter.

**Strengths:**

The paper presents an interesting approach by evaluating the capability of transformer models to mimic the inference process in linear dynamical systems, typically addressed by the Kalman filter. Through empirical analysis, the authors demonstrate how closely the transformer model’s estimation aligns with that of the Kalman filter. The equations presented are logical and are a natural extension of Akyürek et al. (2022) to the Kalman filter algorithm.

**Weaknesses:**

While the paper explores an extension of Akyürek et al. (2022) to linear dynamical systems, it lacks clarity in its motivation, and the overall presentation could be significantly improved. Also, the extension seems logical but limited as Akyürek et al. (2022) has all the parts that is required to implement the model for any linear model. I have merged the weakness and question sections below:

1.	The paper does not explicitly address positional embeddings, which are critical in transformer architectures. Could the authors clarify how positional embeddings were managed in the experiments?
2.	In Figure 1 (among others), why is the error unexpectedly low at a context length of zero? What is the intuition here?
3.	Extending this study to non-linear dynamical systems would add depth and novelty, as demonstrating in-context learning for non-linear cases could yield valuable insights.
4.	As non-linear dynamical systems are considered, comparing the in-context learning performance with algorithms like EKF, UKF, and Gaussian Variational Inference would be beneficial.
5.	Scalability with larger datasets, e.g., up to $10^6$ data points, is an important question. Could low-rank or sparse matrix representations (e.g., in Eq. (20)) help in handling large-scale data efficiently?
6.	In line 481, 'sufficiently long context' is mentioned without specifics. Could the authors provide guidance on the required context length for different datasets or an analysis of its impact?
7.	In Eq. (1), why is normalization omitted?
8.	The presentation of Algorithm 1 is convoluted. Consider using pseudocode instead of describing the operations in prose for better readability.

Some minor suggestions/typos:

1.	Extra bracket in Eq. (1).
2.	L131: $R$ is used to denote both covariance and the real numbers $R^n$.
3.	In Eq (21), $T_\theta$ is undefined.
4.	L248: consider replacing $j$ with $J$ in Div.
5.	Enhance readability by using bold symbols for matrices, $\mathbb{R}$ for real numbers, etc.

**Questions:**

Kindly check the weakness section above.

---

> ### Author Response · Authors · 2024-11-25
>
> Dear Reviewers,
>
> We would like to thank you for your time and effort, and appreciate your constructive feedback.
> Please note that in response to your questions, we significantly expanded the appendix. In
> particular, we added:
> (1) Appendix C, which presents experiments involving control inputs;
> (2) Appendix D, which presents (i) justification of the choice of the SGD learning rate, (ii) MSE
> evaluated with respect to the ground truth (iii) results across varied state dimensions, and
> (iv) an illustration of the performance on out-of-sample parameters;
> (3) Appendix E, which presents preliminary investigation of estimating states in non-linear systems.
>
> Finally, please find detailed responses to your concerns and questions in the text below.
>
> Sincerely,
> Authors
>
> $ \textbf{Question 1:}$ "The paper does not explicitly address positional embeddings, which are critical in transformer architectures. Could the authors clarify how positional embeddings were managed in the experiments?"
>
> $\textbf{Answer:}$ We keep the default positional embedding used in the standard GPT 2 configuration. Thanks for pointing this out, we will include the clarification in the manuscript.
>
> ${\textbf{Question 2}:}$ "In Figure 1 (among others), why is the error unexpectedly low at a context length of zero? What is the intuition here?"
>
> $\textbf{Answer:}$ Please note that the context lengths in our experiments start at 2. This (and several other) figure(s) show the mean-squared prediction difference (MSPD) between the transformer and Kalman filter (as well as between other algorithms and Kalman filter). At low context lengths, both Kalman filter and the transformer achieve similar MSE evaluated with respect to the ground truth -- the low value in the figure simply indicates proximity in the achieved MSEs. Perhaps the newly added Fig. 7 in Appendix D.3 might provide additional intuition -- as can be seen there, the raw MSE is in fact high when the context length is small, and reduces as the context length increases.
>
> ${\textbf{Question 3}:}$ "Extending this study to non-linear dynamical systems would add depth and novelty, as demonstrating in-context learning for non-linear cases could yield valuable insights. As non-linear dynamical systems are considered, comparing the in-context learning performance with algorithms like EKF, UKF, and Gaussian Variational Inference would be beneficial."
>
> $\textbf{Answer:}$ This is indeed a very interesting question, and one we started working on soon after submitting the manuscript to ICLR! The preliminary results for the non-linear system with state equation $x_{k+1}=tanh(x_k)+q_t$ are presented in Appendix E; the obtained MSPD suggests transformer's ability to in-context learn to emulate the Extended Kalman Filter.
>
> $\textbf{Question 4:}$ " In Eq. (1), why is normalization omitted?"
>
> $\textbf{Answer:}$ Please note that layer normalization takes place in equation 3 ($\lambda()$).
>
>
> $\textbf{Question 5:}$ "Scalability with larger datasets, e.g., up to
> $10^6$ data points, is an important question. Could low-rank or sparse matrix representations (e.g., in Eq. (20)) help in handling large-scale data efficiently?"
>
> $\textbf{Answer:}$ Another interesting question. We speculate that it might, although to what extent and how precisely would one exploit special structure will depend on the form of the measurement matrices.
>
>  $\textbf{Question 6:}$ "In line 481, 'sufficiently long context' is mentioned without specifics. Could the authors provide guidance on the required context length for different datasets or an analysis of its impact?"
>
> $ \textbf{Answer:} $ In our experiments, we have observed that for the context lengths that are thrice as long as
> the dimension of the state vector, transformer starts closely emulating Kalman filter and
> achieving low mean-square state estimation error with the MSPD between the Kalman Filter and the Transformer shrinking to less than $5 \\%$ of the peak MSPD.

---

> > ### Comment · Reviewer_AMcU · 2024-11-26
> >
> > Thanks for the response and the extended appendix. However, I still believe more investigation is required related to non-linear SSMs, and they should be part of the main paper as that adds to the novelty of the paper. In addition, the paper is missing a thorough literature review, as reviewer 2Vfm also pointed out. I will keep my score.

---

> > > ### Author Response · Authors · 2024-12-04
> > >
> > > Dear Reviewers,
> > >
> > > Following the received feedback, we made further modification to the manuscript.
> > > In particular, we modified Appendix D.1 to specify line-by-line how the Kalman filter
> > > steps can be expressed in terms of operations that are implementable using the
> > > RAW operator. Furthermore, Appendix D.5 now includes the results for the case
> > > when $h\sim \mathcal{U}[0,3]$ and $h\sim \mathcal{U}[1,4]$. Lastly, Introduction
> > > now highlights differences between the references brought up by Reviewer 2 and
> > > our work.
> > >
> > > Sincerely, Authors

---

### Official Review · Reviewer_cvti · 2024-11-03

**Soundness:** 3
**Presentation:** 3
**Contribution:** 2
**Rating:** 6
**Confidence:** 2

**Summary:**

The paper investigates whether transformer models can learn the behavior of linear dynamical systems via in context learning. Their observations suggest that the transformer emulates the behavior of the Kalman filter which is a statistically optimal estimate of the state, assuming that the system is linear.  The authors explore whether transformers can predict hidden states and observations within state-space models where the system parameters and observations are provided as context. Interestingly, the model maintains robustness even when some parameters, like state transition and noise covariance matrices, are missing.

**Strengths:**

1. The authors provide a fresh perspective of predicting state linear time invariant dynamical systems with transformers and in context learning.
2. The method is well presented
3. The empirical results verify the claim.

**Weaknesses:**

1. While the formulation seems sound, I think it lacks justification as how the context is being setup.  Specifically, can authors present other viable options and have a discussion around these?
2. I prefer there be a short section on Kalman Filters in the background section.
3. How does the method presented extend to non-linear systems?
4. What is the computation requirement?
5. Can authors discuss some viable real life use case?
6. Can there be an ablation study on context length, model size?

**Questions:**

Please see above

---

> ### Author Response · Authors · 2024-11-25
>
> Dear Reviewers,
>
> We would like to thank you for your time and effort, and appreciate your constructive feedback.
> Please note that in response to your questions, we significantly expanded the appendix. In
> particular, we added:
> (1) Appendix C, which presents experiments involving control inputs;
> (2) Appendix D, which presents (i) justification of the choice of the SGD learning rate, (ii) MSE
> evaluated with respect to the ground truth (iii) results across varied state dimensions, and
> (iv) an illustration of the performance on out-of-sample parameters;
> (3) Appendix E, which presents preliminary investigation of estimating states in non-linear systems.
>
> Finally, please find detailed responses to your concerns and questions in the text below.
>
> Sincerely,
> Authors
>
> $\textbf{Question 1}$ "While the formulation seems sound, I think it lacks justification as how the context is being setup. Specifically, can authors present other viable options and have a discussion around these?"
>
> $\textbf{Answer:}$ The setup of the context is due to the requirement that the estimates are formed causally, i.e., that the estimates of present and future states/outputs are formed using past and present measurements. For example, one wants to train the transformer to predict, for $n=1,2,...,N$, $\hat{y}_n$ given $F$, $Q$, $R$, $H_1,y_1, H_2, y_2, \dots, H_n$. One can think of trivial rearrangements of the components of the context that accomplish the same (e.g., group all $H_i$'s and provide them as context ahead of $y_i$'s) but it is unclear if such rearrangements could be beneficial.
>
> $\textbf{Question 2}$ "How does the method presented extend to non-linear systems?"
>
> $\textbf{Answer:}$ We are currently pursuing an extension of the results to non-linear dynamical systems, and are reporting preliminary results for the state transition equation $x_{k+1}=tanh(x_k)+q_t$ (specifically, the MSPD achieved against the Extended Kalman Filter) in Appendix E.
>
> $\textbf{Question 3}$ "What is the computation requirement?"
>
> $\textbf{Answer:}$ Inference for a context length of 40 requires $5.15$ GMACs (or) $10.3$ GFLOPs, which on RTX 3080 would take $345 \mu s$ at peak performance.
>
> $\textbf{Question 4}$"Can authors discuss some viable real life use case?"
>
> $\textbf{Answer:}$ The problem of estimating unobservable states of a dynamical system arises in, e.g., object tracking (radar systems, self-driving vehicles), channel estimation in communication systems, robotics, etc. The practical settings where the system model parameters are partially missing are particularly challenging. In our paper, we demonstrate the transformer's ability to succeed in emulating Kalman filter even when such crucial information is withheld; note that in our simulations the Kalman filter is actually provided the information that is withheld from the transformer.
>
> $\textbf{Question 5}$ "Can there be an ablation study on context length, model size?"
>
> $\textbf{Answer:}$ We appreciate the reviewer's comment. Please note that our evaluation experiments explore the relationship between MSPD and increasing context length; therefore, ablation studies on context length are already included. As for the model size, the short duration of the rebuttal period unfortunately render additional ablation studies challenging.

---

> ### Comment · Reviewer_cvti · 2024-11-25
>
> Dear authors,
>
> Thank you for your comment. My score remains the same

---

### Official Review · Reviewer_F7Fo · 2024-11-04

**Soundness:** 3
**Presentation:** 3
**Contribution:** 3
**Rating:** 5
**Confidence:** 4

**Summary:**

This paper demonstrates that ICL in a transformer can mimic a Kalman filter. Theoretical results show a hand-coded transformer can emulate the internal calculations of the KF algorithm. Experiments training a transformer to predict a linear-Gaussian SSM show its behavior is close to the optimal KF predictions.

**Strengths:**

This is an advance on the recent literature showing how transformers can implement incremental learning algorithms in their forward pass, extending this work to the KF.

The robustness to missing hyperparams is impressive and suggests the trained transformer is doing something more than the theoretical hand-coded one. This merits further investigation, especially since filtering with unknown hyperparams is still an active area of research.

**Weaknesses:**

The theoretical result is a relatively straightforward corollary of Akyürek et al. (2022), just needing to show the KF can be reduced to operations of the form in (11)-(12).

The paper shows the KF can be composed from certain elemental operations and that a transformer can implement those operations, but these two facts need to be put together. It’s nontrivial to arrange multiple operations within the layers of one network. In particular, the required number of layers seems to grow linearly with the sequence length, since the calculations must proceed serially from t=1 to t=N. This contrasts with the transformer in the experiments which learns long contexts using a fixed depth.

The theoretical analysis makes no reference to positional embeddings, but (unless I'm wrong) these are essential to implementing the elemental operations on page 5, and for implementing KF calculations sequentially as noted above.

**Questions:**

Are there implications for ICL in other domains, including LLMs?

(1): shapes don’t agree. The correct expression reverses the order of the two main terms and swaps $W^Q$ and $W^K$.

Line 243: $n+1$ should be $n$ (thrice)

p 5: The four operations could be written more succinctly mathematically, e.g. $G^{l}[K]=G^{l-1}[I]G^{l-1}[J]$.

---

> ### Author Response · Authors · 2024-11-25
>
> Dear Reviewers,
>
> We would like to thank you for your time and effort, and appreciate your constructive feedback.
> Please note that in response to your questions, we significantly expanded the appendix. In
> particular, we added:
> (1) Appendix C, which presents experiments involving control inputs;
> (2) Appendix D, which presents (i) justification of the choice of the SGD learning rate, (ii) MSE
> evaluated with respect to the ground truth (iii) results across varied state dimensions, and
> (iv) an illustration of the performance on out-of-sample parameters;
> (3) Appendix E, which presents preliminary investigation of estimating states in non-linear systems.
>
> Finally, please find detailed responses to your concerns and questions in the text below.
>
> Sincerely,
> Authors
>
> $\textbf{Question 1: }$ "Are there implications for ICL in other domains, including LLMs?"
>
> $\textbf{Answer:}$ Since our focus is on estimating states of dynamical systems, the most immediate implication is on the Bayesian filtering problem that arises in non-linear dynamical systems. Some of the first steps in that direction are now reported in newly added Appendix E.

---

> > ### Comment · Reviewer_F7Fo · 2024-12-02
> >
> > Thank you for your reply and the additional work in the revision. My first two main concerns don't seem to have been addressed. I'm also persuaded by comments by other reviewers about theoretical derivations and comparing to prior work. Apologies but given my reassessment I will lower my score to 5.

---

> > > ### Author Response · Authors · 2024-12-03
> > >
> > > Dear Reviewers,
> > >
> > > Following the received feedback, we made further modification to the manuscript.
> > > In particular, we modified Appendix D.1 to specify line-by-line how the Kalman filter
> > > steps can be expressed in terms of operations that are implementable using the
> > > RAW operator. Furthermore, Appendix D.5 now includes the results for the case
> > > when $h\sim \mathcal{U}[0,3]$ and $h\sim \mathcal{U}[1,4]$. Lastly, Introduction
> > > now highlights differences between the references brought up by Reviewer 2 and
> > > our work.

---

### Official Review · Reviewer_2Vfm · 2024-11-04

**Soundness:** 2
**Presentation:** 1
**Contribution:** 2
**Rating:** 3
**Confidence:** 4

**Summary:**

The paper investigates the ability of a Transformer to learn a Kalman filter for estimation of a linear dynamical system using a tailored context. The paper details the construction of such a context and provides detailed experiments to validate state and output estimates of the transformer against several classical estimation techniques.

**Strengths:**

The paper studies a relevant and interesting connection between control/estimation theory and deep learning, which fits well in the contemporary research focus. The paper is overall well-written and the mathematical derivations are correct. It sufficiently extends existing literature on the transformer’s ability to represent various algorithms, e.g. SGD, via in-context learning.

**Weaknesses:**

However, the paper has several weaknesses that render the paper reading like a draft to an interesting and promising paper.

First, the paper only contains few references and does not accurately position itself in existing literature.  The paper lacks a sufficient discussion on recent works connecting the transformer architecture to linear state space models, see e.g. [1.2]. The paper also ignores work on learning the Kalman filter not related to transformers, see e.g. [3,4]. These works should at least be referenced as related work and ideally be discussed and compared against. Additionally, there are missing references for the GPT2 model [5] and the Adam optimization algorithm [6].

[1] Dao & Gu, “Transformers are SSMs: Generalized Models and Efficient Algorithms Through Structured State Space Duality”, ICML 2024

[2] Sieber et al., “Understanding the differences in Foundation Models: Attention, State Space Models, and Recurrent Neural Networks”, NeurIPS 2024

[3] Tsiamis et al., “Sample Complexity of Kalman Filtering for Unknown Systems”, PMLR 2020

[4] Krishnan et al., “Deep Kalman Filters”, 2015

[5] Radford et al., “Language Models are Unsupervised Multitask Learners”, 2019

[6] Kingma & Ba, “Adam: A Method for Stochastic Optimization”, 2014

Second, the mathematical and experimental presentation is not sufficient. Regarding mathematical derivations, almost all details are not presented, which would significantly improve the paper and would allow the reader to more easily follow the arguments. For example, there should be more background on the RAW operator (11 -12), which can be given in the appendix to make the paper self-contained.  The index matrices I_{B1}, …, I_B{8}, I_F, etc. should be stated in the appendix. Currently, these matrices are assumed to be inferred by the reader, which severely hampers the readability. Additionally, the derivation of the contexts (20), (28) should be derived or the derivation at least be discussed in more detail. The same goes for Algorithm 1 (and Algorithm 2 in the appendix); there is no derivation or explanation to how these are obtained. The statement that eigenvalues on the unit circle are unstable is false according to standard definitions of stability. These eigenvalues are marginally stable, since the state does neither converge nor diverge in this case. Observability and Detectability of the linear system is not discussed at all. However, these two notions are important in this setting (F, H are randomly generated), since the ability to observe or detect the state from the specific F, H is important for the resulting performance of the Kalman filter and ultimately the transformer. Regarding this point, the presentation of the results is insufficient. Currently, only the relative performance (i.e. transformer compared to Kalman filter etc.) is stated, but the results would be much stronger if the actual ground truth (which is available) is reported, i.e., state the ground truth state/output and then the error of Kalman filter, transformer, and the other estimation algorithms w.r.t. the ground truth. Additionally, the plots (e.g. Figure 1) should not be clipped, but maybe split in two: one plot showing the full plot (e.g. on a logarithmic scale) highlighting the performance difference and a second plot showing the detailed differences between the better performing algorithms. Finally, the used codebase (Garg, 2022) should be hyperlinked and all the experimental details should be provided; currently this is insufficient.

Third, in my opinion the title is misleading. The title insinuates that the behavior, i.e., the dynamical representation of a linear system, is learned. However, the transformer actually learns to estimate the hidden state or output of a linear system. These two things are not equivalent. Additionally, the appendix is only used to give an extension of the method to dual Kalman filters instead of providing additional details of the main method.

Lastly, there are several typos in the equations and grammar mistakes (mainly prepositions), some of which are listed below. However, the authors should make an effort to carefully check the full paper beyond the following list.

-	In (9) and (10), the inverse should be a pseudoinverse, since in general these inverses do not exist.
-	In (4) the matrix F is time-varying, but later on all derivations are done for time-invariant F. Since it is assumed that F is time-invariant, why not state (4) for time-invariant F? Additionally, mention why Q, R are not time-varying here; in general they are time-varying.
-	Line 259: \mathcal{A}_append should be \mathcal{A}_prepend
-	Line 177: comprising of examples
-	Line 201: assume a time-invariant; For simplicity of presentation (no “the”)
-	Line 202: we consider scalar measurements at first.
-	Lines 217-218: the computationally intensive matrix inversion in (15) simplifies to a simple scalar division
-	Line 226: how closely a transformer can mimic

**Questions:**

-	Does the in-context learning also work for time-varying F? I assume the derivations would only differ slightly, so this is more of an experimental question. Also did you consider time-varying Q, R?
-	Regarding the performance gap between Strategies 1 and 2: Is the reason for this gap the stability of the system or is it the specific sampling of F? You could check this be removing alpha=1 in Strategy 1 (see below).
-	Related to the question above, why do you include alpha = 1 in Strategy 1? You could just vary alpha in [0,1).
-	Have you considered other learning rates for SGD? Your results suggest that by tuning the learning rate the performance of the SGD estimation could be close to Kalman filtering.

---

> ### Author Response · Authors · 2024-11-25
>
> Dear Reviewers,
>
> We would like to thank you for your time and effort, and appreciate your constructive feedback.
> Please note that in response to your questions, we significantly expanded the appendix. In
> particular, we added:
> (1) Appendix C, which presents experiments involving control inputs;
> (2) Appendix D, which presents (i) justification of the choice of the SGD learning rate, (ii) MSE
> evaluated with respect to the ground truth (iii) results across varied state dimensions, and
> (iv) an illustration of the performance on out-of-sample parameters;
> (3) Appendix E, which presents preliminary investigation of estimating states in non-linear systems.
>
> Finally, please find detailed responses to your concerns and questions in the text below.
>
> Sincerely,
> Authors
>
> $\textbf{Question 1: }$ "Does the in-context learning also work for time-varying F? I assume the derivations would only differ slightly, so this is more of an experimental question. Also did you consider time-varying Q, R?"
>
> $\textbf{Answer:}$ While the theoretical analysis can indeed be readily extended to time varying $F$, $Q$, and $R$, experiments may not be as readily scalable since one needs to provide $F_1,...,F_N, Q_1,..., Q_N$ as context. For example, for the context length $40$ and the state dimension $8$, time varying $F$ would necessitate appending $40$ $8\times 8$ matrices as context. It is important to note that for most practical systems, the state transition matrix and the noise covariances change, if at all, at a much lower rate than the sensing rate.
>
> $\textbf{Question 2:}$  "Regarding the performance gap between Strategies 1 and 2: Is the reason for this gap the stability of the system or is it the specific sampling of F? You could check this be removing alpha=1 in Strategy 1 (see below).
> Related to the question above, why do you include alpha = 1 in Strategy 1? You could just vary alpha in [0,1)."
>
> $\textbf{Answer:}$ Please note that $F$ is stable under both Strategy 1 and 2. The smaller MSPD for the second strategy is likely due to the rapidly diminishing norm of the state over time. For example, if we set process noise to 0, the norm of the state is $\|x_N\|\leq\|F^N\| \|x_0\|$ by the Cauchy-Schwartz inequality, while $\|F^N\|$ is much smaller for Strategy 2 than for Strategy 1.
>
>
> $\textbf{Question 3:}$ "Have you considered other learning rates for SGD? Your results suggest that by tuning the learning rate the performance of the SGD estimation could be close to Kalman filtering."
>
> $\textbf{Answer:}$ Yes, we performed a grid search over varying SGD learning rates with a fixed context length of 40 and selected two rates resulting in the lowest MSPD against the transformer's output. In response to the reviewer's question, we created Appendix D.2 which provides details regarding the choice of the learning rate.

---

> > ### Comment · Reviewer_2Vfm · 2024-11-26
> >
> > Dear authors,
> >
> > Thank you for your answers and considering my suggestions. While you address some of my concerns, some of the weaknesses outlined in the review are not addressed at all:
> > 1. In the revised pdf, there is still no sufficient discussion on the state of the art beyond the immediate related work. Additionally, the missing references are not added.
> > 1. As also pointed out by Reviewer 1cwb, the mathematical/theoretical derivations are not present in the paper; as are other theoretical considerations I was pointing out (e.g. Observability/Detectability). The main difference between the initial submission and the revised version are the additional experiments in the appendix.
> > 1. Regarding the answer to Question 2: I don't understand why $F$ should be strictly stable for both strategies, when you explicitly state in the paper that $F$ for Strategy 1 might by marginally stable (eigenvalues equal to 1). Could you clarify this? Also, if $F$ has eigenvalues equal to 1, the inequality is strict which warrants some discussion in the paper (assuming the norms in your answer are 2-norms).
> >
> > Therefore, I will remain with my score.

---

> > > ### Author Response · Authors · 2024-12-04
> > >
> > > Dear Reviewers,
> > >
> > > Following the received feedback, we made further modification to the manuscript.
> > > In particular, we modified Appendix D.1 to specify line-by-line how the Kalman filter
> > > steps can be expressed in terms of operations that are implementable using the
> > > RAW operator. Furthermore, Appendix D.5 now includes the results for the case
> > > when $h\sim \mathcal{U}[0,3]$ and $h\sim \mathcal{U}[1,4]$. Lastly, Introduction
> > > now highlights differences between the references brought up by Reviewer 2 and
> > > our work.
> > >
> > > Finally, please find detailed responses to your concerns and questions in the text below.
> > >
> > > Sincerely, Authors
> > >
> > > 1.  Please note that, as we now state in the Introduction, the references the reviewer brought up have no close relation to our work. Specifically, works of Dao\& Gu (2024) and Sieber et al. (2024) have no connection to the study of state estimation, filtering, or in-context learning. We cite them nevertheless, as the reviewer required. Krishnan et al. (2015) is closer to KalmanNets(Revachetal.,2021) in so far as it replaces linear transformations with non-linear transformations parameterized by neural nets. They fit a generative model to a sequence of observations and actions (control inputs). They do not evaluate their framework on arbitrary state space models, and their work is not related to in-context learning that is the focus of our work. Lastly, Tsiamis et al. (2020) pertain more to system identification problem where they obtain coarse estimates of the parameters of the state space model and design a filter based on the approximate parameters to estimate the state subject to certain robustness guarantees.  However, their work is not related to in-context learning or Deep learning in general.
> > >
> > >
> > > 2. When the measurement matrix is a row vector which is the case for almost all of our experiments, the system is not necessarily observable. But Kalman Filtering still provides the minimum mean squared error among all the linear estimators of the state regardless of the observability.
> > >
> > > 3. When the system is not stable, there are initial states for which the magnitude of the state increases exponentially. For such systems, the error covariance of one-step prediction by the Kalman filter is large, and this causes issues in the training of the transformer as well.  For marginally stable systems, the eigenvalues lie on the unit circle, and consequently barring noise, the magnitude of the state is the same as that of the initial state. In the paper we do not claim that transformer cannot learn marginally stable systems. We rather make the case that for transformer to learn one-step prediction for a marginally stable system, we need to implement a curriculum in which we increase alpha from 0 to 1. Lastly, to clarify our rebuttal to question 2, F is stable for both strategy 1 and 2 during the evaluation as we sample $\alpha$ from $\mathcal{U}[0,1]$

---

### Official Review · Reviewer_1cwb · 2024-11-04

**Soundness:** 1
**Presentation:** 2
**Contribution:** 3
**Rating:** 3
**Confidence:** 3

**Summary:**

The paper studies the capability of transformer-based models to implicitly learn the closed-form update equations of the Kalman filter. The paper lists an algorithm for implementing the Kalman filter equations using operations that can readily be implemented through appropriate weights of a transformer architecture. The claims are empirically investigated by comparing mean squared prediction difference between transformer’s in-context learning and Kalman filter vs SGD, ridge regression, and ordinary-least-squares.

**Strengths:**

The investigation of how modern deep learning architectures can learn complex behaviour such as Kalman filtering as an _in-context_ task is highly significant to ICLR. Whereas Goel et al. showed that transformers can implement Kalman filters up to a (small, bounded) additive error, when trained for a specific dynamical system, the present work aims to demonstrate that transformers can learn not just a _specific_ Kalman filter, but more generally the Kalman filter equations such that it can in-context learn any dynamical system whose parameters are provided in the context.

**Weaknesses:**

The paper does not provide a derivation of its theoretical claims; the paper simply asserts that Algorithm 1 provides a transformer-appropriate implementation of the Kalman update equations. Contrast this to e.g. Akyürek et al. (ICLR 2023; one of the key citations in this paper) who include a detailed derivation of their results in the appendix.

Regarding empirical evaluation, this paper also falls significantly short of the level set by previous investigations into the in-context learning capabilities of transformers. It considers only a proxy metric on a test set that is generated from the same distribution as the training set. In contrast, Garg et al. (NeurIPS 2022; again a key citation in this paper) investigate the out-of-distribution generalization as well as how performance depends on model capacity and problem dimensionality. Both of those would be just as important in the present study: demonstrating that the learned transformer performs as well out-of-distribution would be much stronger evidence for actually having learnt the Kalman filter than solely comparing the difference in prediction to a Kalman filter on dynamical systems from within the training distribution.

**Questions:**

Can you motivate your choices of setup, architecture, training schedule etc. (beginning of section 4)? Why did you choose specifically your Strategies 1 and 2 for generating $F$?

Maximum marginal noise is 0.025: how much noise is this relative to the scale of dynamics (low-noise, high-noise, …)?

What does it mean that MSPD peaks when context length approximately equals state dimensionality? Does this hold for multiple state dimensions? If so, a discussion would be useful (and a visualization of the state dimension as a vertical line in Fig. 1).

## Clarity questions

1. line 136: what are the model parameters you are referring to?
2. line 151: what is the convergence criterion? It seems from the equation you simply run a fixed number of $N$ steps.
3. line 202: what do you mean by _causal_ linear estimator?
4. line 205: what are $\hat{x}_0^+$ and $\hat{P}_0^+$ (not introduced)?
5. line 230: in eq. (20), why do you provide $Q$, but not $R$ (also in eq. (28))? Corollary, in line 417 you discuss omitting $R$ and $Q$ from context, which would be at odds with $R$ never being in the context in the first place.
6. line 235: in eq. (21), what is $T_\theta$? (also in eq. (30))
7. lines 311–317: are $Q$, $R$ newly sampled for each example (like $x_0$, $H$)?
8. line 320: what is $\phi$, can you clarify what you are doing here?
9. lines 352–354: from step 50000 onwards, $\alpha=1$, does this mean the transformer’s loss no longer decreases for the second half of training?
10. line 367: is the initialization for the Kalman filter comparable to the initialization you use in the transformer model?

Please carefully check your references. For example, “What learning algorithm is in-context learning? Investigations with linear models” by Akyürek et al. has been published at ICLR 2023 (as is clear from the arXiv entry!), similarly for other citations only listed as arXiv preprints.

## Minor nits
- Eq. (1): should be $\operatorname{Softmax}$ instead of $Softmax$
- Section 2.2: Please use `\mathbb{R}` ($\mathbb{R}$) for the space of real numbers instead of just $R$ (which is ambiguous with your notation for covariance of measurement noise)
- Below eq. (5), I’m assuming you mean that $q_t$ and $r_t$ are Gaussian-distributed noise; please be explicit about $q_t \sim \mathrm{N}(0, Q)$, $r_t \sim \mathrm{N}(0, R)$: white noise is not necessarily Gaussian, only describes independence over $t$
- line 184: would be helpful to explain what is the _RAW_ operator, at the very least spell out that it stands for Read–Arithmetic–Write as described in Akyürek et al. (2023)
- Below eq. (21), it’d be useful to explicitly state you use 0-based indexing (arguably default in programming, but in mathematical notation I would expect 1-based indexing by default if not stated otherwise!)
- line 413: “eingenvalues” should be “eigenvalues”

---

> ### Author Response · Authors · 2024-11-25
>
> Dear Reviewers,
>
> We would like to thank you for your time and effort, and appreciate your constructive feedback.
> Please note that in response to your questions, we significantly expanded the appendix. In
> particular, we added:
> (1) Appendix C, which presents experiments involving control inputs;
> (2) Appendix D, which presents (i) justification of the choice of the SGD learning rate, (ii) MSE
> evaluated with respect to the ground truth (iii) results across varied state dimensions, and
> (iv) an illustration of the performance on out-of-sample parameters;
> (3) Appendix E, which presents preliminary investigation of estimating states in non-linear systems.
>
> Finally, please find detailed responses to your concerns and questions in the text below.
>
> Sincerely,
> Authors
>
> $\textbf{Question:}$ "Can you motivate your choices of setup, architecture, training schedule etc. (beginning of section 4)? Why did you choose specifically your Strategies 1 and 2 for generating $F$
> ?"
>
> $\textbf{Answer:}$ The parameters of the state-space model are sampled in a way to conform with theoretical restrictions. For example, the process covariance matrix needs to be symmetric and  positive semi-definite, and consequently, the eigenvalues are sampled from a uniform distribution (rather than, e.g., a normal distribution). The same holds for the variance of the measurement noise which needs to be positive. As for the choice of $F$, Strategy 1 and 2 are convenient approaches to randomly generating $F$ that leads to a stable system. As for the architecture, we use GPT 2 model to maintain consistency and allow fair comparison with the prior work [Akyurel et al.].
>
>
> $\textbf{Question:}$ "Maximum marginal noise is 0.025: how much noise is this relative to the scale of dynamics (low-noise, high-noise, …)?"
>
> $\textbf{Answer:}$ The power of the noiseless measurements is 0.57 (evaluated empirically); since the power of the noise is 0.025, the resulting SNR is 13.57dB.
>
> $\textbf{Question:}$ "What does it mean that MSPD peaks when context length approximately equals state dimensionality? Does this hold for multiple state dimensions? If so, a discussion would be useful (and a visualization of the state dimension as a vertical line in Fig. 1)."
>
> $\textbf{Answer:}$ It simply means that we observe the largest gap between the MSE performance of the transformer and Kalman filter when the context length approximately equals state dimension. We will gladly add a brief discussion on this to the manuscript.
>
> $\textbf{Derivation for the Theoretical Claims:}$ The claims are supported via a proof by construction -- specifically, by the construction of Algorithms 1 and 2 which utilize operations readily performed by a transformer to run Kalman filter recursion. Building on the results of Akyurek et al., we in fact do show in the paper that transformers can implement these essential operations.
>
> ${\textbf{Out-of-distribution generalization.}}$ Please note that in our presented experiments, the noise during testing is generated from a distribution whose parameters are randomized (in particular, variance of the noise is generated uniformly at random). Moreover, in response to the reviewer's comment we evaluated transformer's performance when the system's parameters are generated from a distribution different from the one seen during the training. The results, now included in the appendix section D.5, clearly demonstrate transformer's ability to emulate Kalman filter on out-of-distribution data.
>
> $\textbf{Clarity Question 1 }$ "line 136: what are the model parameters you are referring to?"
>
> $\textbf{Answer:}$ We are referring to the parameters of the underlying system model.
>
> $\textbf{Clarity Question 2: }$ "line 151: what is the convergence criterion? It seems from the equation you simply run a fixed number of
>  $N$ steps"
>
>  $\textbf{Answer:}$ Here we run SGD on all available context examples and refer to the convergence of the training loss.
>
> $\textbf{Clarity Question 3: }$ "line 202: what do you mean by causal linear estimator?"
>
> $\textbf{Answer: }$ Causal linear estimators provide estimates of the current or future states/observations by forming weighted linear combinations of the past and current measurements.
>
> $\textbf{Clarity Question 4: }$ "line 205: what are
>  $x_0^{+}$ and $P_0^{+}$
>  (not introduced)?"
>
>  $\textbf{Answer:}$ In our simulations, $x_0^{+}=0$ and $P_0^{+}=I$. Thanks for pointing out the omission, we will specify these values in the manuscript.

---

> > ### Comment · Reviewer_1cwb · 2024-11-25
> >
> > Thanks for the answers, but I remain with my score, as I don't think the updated manuscript sufficiently addresses my key concerns:
> >
> > Theoretical derivation - Akyürek et al carefully guide the reader through all the steps necessary to follow the construction. I would like to see the same for this paper and how it recovers the Kalman filter equations.
> >
> > Out-of-sample experiments - U[-5/4, 5/4] is within the bulk of N(0, 1) so this is very much within distribution.
> >
> > Questions (including clarity questions) - I would like to see the answers to my questions reflected within the updated manuscript.
> >
> > Architecture - I did not mean just "GPT2 style decoder-only", but all aspects such as layer sizes. Akyürek et al. do hyperparameter optimization based on validation loss, so I was wondering how you chose the values you used.
> >
> > "What does it mean that MSPD peaks" - I meant this in the "why" sense: what is the underlying reason for this pattern ?
> >
> > Follow-up questions:
> >
> > "[we] refer to the convergence of the training loss" - yes, but what is your criterion for convergence?

---

> > > ### Comment · Reviewer_1cwb · 2024-11-26
> > > **For future reference...**
> > >
> > > Just for future reference, you can use markdown in openreview, so instead of `$\textbf{Clarity Question 6:}$` (rendered as $\textbf{Clarity Question 6:}$, which cannot be copy&pasted) you can simply write `**Clarity Question 6:**` (rendered as **Clarity Question 6:**, which _can_ be copy&pasted).
> > >
> > > The title of the paper (in the openreview meta data) should NOT be in all-caps.

---

> ### Author Response · Authors · 2024-11-25
>
> $\textbf{Clarity Question 5: }$ "line 230: in eq. (20), why do you provide
> $Q$, but not $R$
>  (also in eq. (28))? Corollary, in line 417 you discuss omitting
> $R$ and $Q$
>  from context, which would be at odds with $R$
>  never being in the context in the first place."
>
> $\textbf{Answer:}$ We actually do provide $R$, a diagonal matrix, to the model. In particular, when the measurement vector is $m$-dimensional, noise variances $\sigma_1^2$, $\sigma_2^2$,$\dots$, $\sigma_m^2$
> are part of the context provided to the transformer.
>
> $\textbf{Clarity Question 6: }$"line 235: in eq. (21), what is $T_\theta()$
> ? (also in eq. (30))"
>
> $\textbf{Answer:}$ $T_{\theta}()$ represents the output of the transformer.
>
>
> $\textbf{Clarity Question 7: }$ "lines 311–317: are
> $Q$,$R$
>  newly sampled for each example (like $x_0$
> , $H$
> )?"
>
> $\textbf{Answer:}$ Yes, $Q$ and $R$ are indeed sampled anew for every example.
>
> $\textbf{Clarity Question 8: }$ "line 320: what is
> $\phi$, can you clarify what you are doing here?"
>
> $\textbf{Answer:}$ Please note that the eigenvalues of matrix $F$ are in general complex valued and can thus be expressed as $re^{j \phi}$, where $r$ denotes the magnitude and $\phi$ denotes the phase of the said eigenvalue.
>
> $\textbf{Clarity Question 9: }$ "lines 352–354: from step 50000 onwards,
> $\alpha$=1, does this mean the transformer’s loss no longer decreases for the second half of training?"
>
> $\textbf{Answer:}$ The transformer's loss continues to decrease after $50000$ steps (with $\alpha=1$) because of the curriculum learning; one can think of the weights of the transformer learned by the 50000th steps as the initialization for the second phase of training.
>
> $\textbf{Clarity Question 10: }$ "line 367: is the initialization for the Kalman filter comparable to the initialization you use in the transformer model?"
>
> $\textbf{Answer: }$ The Kalman filter is initialized by setting $x_0^{+}=0$ and $P_0^{+}=I$; these values show up in  $\mathcal{A}_{cat}$, which is used for the proof-by-construction arguments, but are not part of the input matrix provided to the transformer in the numerical experiments.

---

> ### Author Response · Authors · 2024-12-04
>
> Dear Reviewers,
>
> Following the received feedback, we made further modification to the manuscript.
> In particular, we modified Appendix D.1 to specify line-by-line how the Kalman filter
> steps can be expressed in terms of operations that are implementable using the
> RAW operator. Furthermore, Appendix D.5 now includes the results for the case
> when $h\sim \mathcal{U}[0,3]$ and $h\sim \mathcal{U}[1,4]$. Lastly, Introduction
> now highlights differences between the references brought up by Reviewer 2 and
> our work.
>
> Finally, please find detailed responses to your concerns and questions in the text below.
>
> Sincerely, Authors
>
> 1. For architecture, we obtained satisfactory results with 16 layered transformer results with 4 heads and a hidden dimension of 512. Resource constraints limited our
> ability to perform further exploration of the hyper-parameters
>
>
> 2. We run SGD for a maximum of 100 epochs. We observe that the training loss converges to a constant value within these many epochs. After running SGD in the context, we predict the output of the query.

---

### Meta-Review · Area_Chair_ECA8 · 2024-12-23

**Metareview:**

This paper explores the in-context learning abilities of transformer models, in particular with regards to learning the Kalman Filter. The authors provide an algorithm for constructing such a context and then they validate their methodology by comparing the transformer approach against classical estimation methods.

This paper is a step forward from previous works, since it is not limited to investigating specific dynamical systems but, in contrast, it is concerned with the actual Kalman Filter equations. This fresh view, is a significant generalization of the motivation to study transformers in-context learning capabilities for dynamical systems.

The exploration also covers the case where system hyperparameters are partially missing, this is an interesting study and it is also impressive to see that the algorithm remains robust.

On the other hand, there were many concerns raised by the reviewers. Almost all reviewers found parts of the theoretical justifications to be either incomplete or unsatisfactorily presented. There also were many clarification questions, hinting towards the fact that the presentation of the paper should be improved. Some reviewers, similarly, found the experimental evaluation to be incomplete, see in particular comments of reviewer 1cwb and 2Vfm.

Overall, this paper has benefitted from the review of five expert reviewers who also remained engaged during the rebuttal period, however there seems to be a consensus that the paper is not ready yet for publication.

**Additional Comments On Reviewer Discussion:**

There were many clarification questions, as well as discussions related to the aspects of the paper that the reviewers consider to be weaknesses - see meta-review. While some progress was made during the rebuttal, it seems that the issues go beyond what can be addressed during the rebuttal period, and a new submission would be needed.

It is indicative that after the first rebuttal phase, every reviewer chose to remain with their score, whereas one of the reviewers actually lowered it.

---

### Decision · Program_Chairs · 2025-01-22

Reject